# Stochastic Process Learning via Operator Flow Matching

**Yaozhong Shi**
California Institute of Technology
yshi5@caltech.edu

**Zachary E. Ross**
California Institute of Technology
zross@caltech.edu

**Domniki Asimaki**
California Institute of Technology
domniki@caltech.edu

**Kamyar Azizzadenesheli**
NVIDIA Corporation
kaazizzad@gmail.com

## Abstract

Expanding on neural operators, we propose a novel framework for stochastic process learning across arbitrary domains. In particular, we develop operator flow matching (OFM) for learning stochastic process priors on function spaces. OFM provides the probability density of the values of any collection of points and enables mathematically tractable functional regression at new points with mean and density estimation. Our method outperforms state-of-the-art models in stochastic process learning, functional regression, and prior learning.

## 1   Introduction

Stochastic processes are foundational to many domains, from functional regression and physics-based data assimilation, to financial markets, geophysics, and black box optimization. Stochastic processes can serve as prior distributions over functions and can provide the density of any finite collection of points. Conventionally, these priors are designed by hand from predefined Gaussian processes (GP) and their variants, and therefore assume that they adequately describe the phenomena of interest. However, many processes modeled in the natural world are not well described by GP, Fig 2. Such models limit the flexibility and generalizability of these stochastic processes in real-world applications, leaving behind significant challenges for more general stochastic process learning (SPL).

In SPL, the prior over the stochastic process is learned from data, i.e., a set of historical point evaluations in past experiments. Learning the prior over the process is crucial for universal functional regression (UFR), which is a recently proposed Bayesian scheme for functional regression and takes GP-regression as a special case when the prior is Gaussian [1]. UFR is important to scientific and engineering domains, including reanalysis, data completion-assimilation, uncertainty quantification, and black box optimization.

In this paper, we introduce a novel operator learning framework termed operator flow matching (OFM) for learning priors over stochastic processes through the joint distribution of any collection of points. To achieve this, we theoretically and empirically generalize marginal optimal transport flow matching [2] to infinite-dimensional function spaces where we map a GP into a prior over function spaces through a flow differential equation. We then derive SPL from the function space derivation and learn a prior over arbitrary sets of points. For SPL, we map any collection of pointwise evaluations of a GP to pointwise evaluations of target functions. This allows us to learn prior distributions over more general stochastic processes, hence enabling sampling values of any collection of points with their associated density and facilitating efficient UFR. We leverage this capability by extending neural operators [3]–designed initially to map functions between infinite-dimensional spaces–to maps

39th Conference on Neural Information Processing Systems (NeurIPS 2025).

between collections of points deploying their functional convergence properties. This serves as the essential architecture block in OFM.

After learning the prior and having access to the densities, OFM can be used for UFR, where given any collection of points of the underlying function, we estimate the posterior mean value of any new collection of points and efficiently sample from their posterior values using stochastic gradient Langevin dynamics (SGLD) [4], i.e., a Gaussian sampling on the input GP space (Fig 1). We show that OFM significantly outperforms state-of-the-art (SOTA) methods, including deep GPs, conditional models, and operator flows (OPFLOW) [1, 5–10].

Generalizing GP-regression to regression over general stochastic process in a practical and implementable way demands a unified framework that spans many key fields. Because readers will have diverse backgrounds and expertise, we provide a set of potential questions and answers in Appendix A to further clarify the development and highlight our contributions.

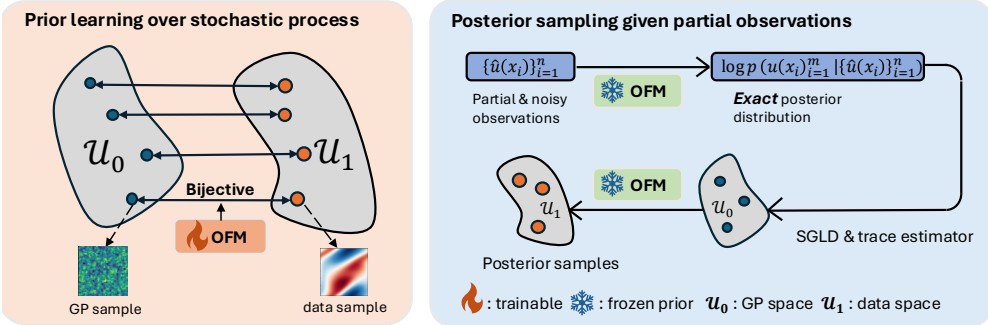

Figure 1: Two-phase strategy for prior learning and posterior sampling. In the prior-learning phase, OFM leverages the marginal optimal-transport path to learn a bijective mapping between a predefined GP and the unknown stochastic process that generates the training data. In the posterior-sampling phase, the learned prior is frozen; given noisy, partial observations, the exact posterior is obtained via Bayes' theorem. SGLD, aided by the Hutchinson trace estimator, then enables efficient and robust sampling.

Lastly, our contributions are summarized as follows:

- **First** work to extend *flow matching / stochastic interpolants / rectified flow* [11–13] to stochastic processes via operator learning. The formulation enables likelihood estimation for function values at any collection of points for the target stochastic process. We also contribute to the development of marginal (dynamic) optimal transport in infinite-dimensional flow matching through optimal coupling and dynamic Kantorovich formulations.
- **First** integration of flow matching with functional regression yields a unified framework for prior and posterior sampling that is applicable to both generation and regression tasks.
- A practical generalization of GP regression that provides the **exact** prior and posterior density over an unknown stochastic process (whether GP or non-GP). In contrast, previous methods such as deep GPs and conditional models work with approximate posteriors. Our method achieves **SOTA** performance in all challenging functional regression tasks.
- This work provides a unified perspective bridging several important fields, which opens new research directions for problems in science and engineering. Additionally, we present extensive ablation and scaling studies that demonstrate the effectiveness of each component in our framework.

## 2 Related Work

**Neural operators.** Neural operators constitute a paradigm in machine learning for learning maps between function spaces, a generalization of conventional neural networks that map between finite dimensional spaces [14, 15]. Among neural operator architectures, Fourier neural operators (FNO) [14] enable convolution in the spectral domain and are effective for operator learning [16–21]. In this work, we use this as our neural operator architecture.

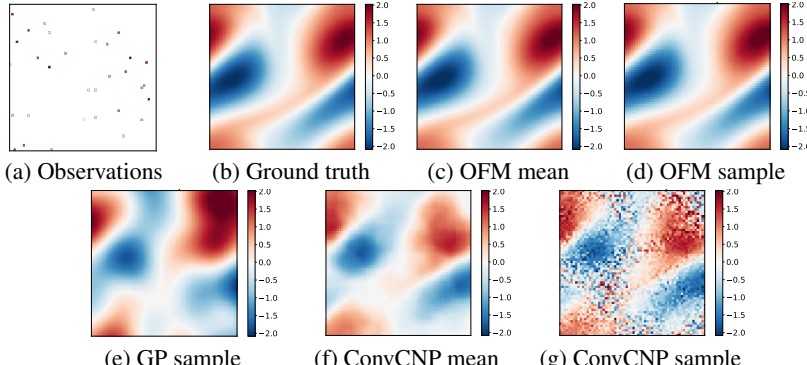

Figure 2: Operator Flow Matching (OFM) regression on Navier-Stokes functional data with resolution $64 \times 64$. (a) 32 random observations (only 0.7%). (b) Ground truth sample. (c) Predicted mean from OFM. (d) One posterior sample from OFM. (e) One posterior sample from the best fitted GP. (f) Predicted mean from ConvCNP. (g) One posterior sample from ConvCNP.

**Direct function samples.** There is a body of work on generative models dedicated to learning distributions over functions, such that direct sampling on the function space is possible. For example, generative adversarial neural operators (GANO) generalize generative adversarial nets on finite dimensional spaces to function spaces [22, 23], yielding a neural operator generative model that maps GP to data functions [3]. Other works in this area have followed the success of diffusion models [24, 25] in finite dimensional spaces, e.g., denoising diffusion operators generalize diffusion models to function spaces by using GP as a means to add noise and use neural operators to learn the score operator on function-valued data [26–28]. Moreover, the same principle has been deployed to generalize flow matching [11] to functional spaces [29], an approach closely related to our work. However, these works on learning generative models on function spaces do not support UFR the way GP-regression does because they (i) focus solely on generating function samples, (ii) do not clarify how to model a stochastic process on point value sequential generation, and (iii) do not provide point evaluation of probability density. In contrast, OFM enables exact density calculation and conditional posterior sampling.

**Stochastic processes.** Earlier works on SPL have focused on hand-tuned methods in the style of GP-regression. In these cases, an expert tunes the GP parameters given a set of experimental samples. More advanced methods rely on deep GPs, in which a network of GPs is stacked on top of each other. The parameters of deep GPs are commonly optimized by minimizing the variational free energy, which serves as a bound on the negative log marginal likelihood. [30, 31]. Deep GPs have limitations in terms of learnability, expressivity, and computational complexity. Warped GPs [32] and transforming GP [33] methods use historical data to learn a pointwise transformation of GP values and achieve on par performance compared to deep GP type methods. The pointwise nature of such approaches limits their generality.

Another line of work proposes learning the conditional distribution of point values, inspired by variational inference and designed for sampling from function spaces; we refer to this line of models as conditional models[1] [7]. This method trains a model to map any collection of points and their values to a vector, used as an input to a decoder that maps any collection of points to their values. The architectures used in these models (including the decoder) are not mathematically consistent as the number of points grows, limiting the approach to finite dimensions. In particular, Convolutional Conditional Neural Processes (ConvCNPs) [6] use convolutional architectures to capture local, translation-invariant structure, but they assume stationarity and struggle with long-range dependencies or highly non-uniform data. The diffusion based variants [34] also use uncorrelated Gaussian noise, and the results do not exist in function spaces [22, 26]. Furthermore, methods based on conditional models are unable to provide density estimation for collections of points, as needed for UFR and SPL in general.

Separately, OPFLOW introduced invertible neural operators that are trained to map any collection of points sampled from a GP to a new collection of points in the data space [1], using the maximum likelihood principle. This method is mathematically consistent as the resolution grows, captures

---

[1]Neural process (NP) is a prominent example of conditional models

the likelihood of any collection of points, and allows for UFR using SGLD. However, similar to normalizing flow [35] methods in finite dimensional domains, the use of invertible deep learning models makes their training a challenge, particularly with regard to expressiveness, as also described in the original OPFLOW work. Finally, we strongly encourage readers to consult Appendix A and Q for an in-depth comparison of stochastic process learning and other generative frameworks.

## 3 Operator Flow Matching

Here, we introduce the problem setting and notations used for OFM in function space. We recommend that readers consult Appendix B–E for a foundational overview of SPL, UFR and related background. Subsequently, we present the framework of OFM, which extends marginal optimal transport flow matching [2] to infinite-dimensional spaces. We further demonstrate the generalization of flow matching to stochastic processes as it is induced from OFM on function spaces. Finally, we illustrate how to evaluate exact and tractable likelihoods for any point evaluation of functions using OFM, making it applicable in the UFR setting.

For a real separable Hilbert space $(\mathcal{H}, \langle \cdot, \cdot \rangle, \|\cdot\|)$, equipped with the Borel $\sigma-$ algebra of measurable sets denoted by $\mathcal{B}(\mathcal{H})$, we introduce two measures on $\mathcal{B}(\mathcal{H})$, $\nu_0$ as the reference measure and $\nu_1$ as the data measure. Consider a function $h_0$ sampled from $\nu_0$, such that $h_0 \sim \nu_0$. For a smooth time-varying infinite dimensional vector field $\mathcal{G}_t : \mathcal{H} \to \mathcal{H}$, we define an ordinary differential equation (ODE)

$$\frac{\partial \Phi_t(h_0)}{\partial t} = \mathcal{G}_t(\Phi_t(h_0)) \tag{1}$$

with initial condition $\Phi_0(h_0) = h_0$, where $h = h_t = \Phi_t(h_0)$ represents a function $h_0$ transported along a vector field from time 0 to time $t$. The diffeomorphism $\Phi_t$ induces a pushforward measure $\mu_t := [\Phi_t]_\sharp(\mu_0)$, with $\mu_0 = \nu_0$, and we refer to $\mu_t$ as the path of probability measure. The goal is to construct a path of probability measure such that at $t = 1$, $\mu_1 \approx \nu_1$. The dynamic relationship between the time varying measure $\mu_t$ and vector field $\mathcal{G}_t$ can be characterized by the continuity equation:

$$\frac{\partial \mu_t}{\partial t} = -\nabla \cdot (\mu_t \mathcal{G}_t) \tag{2}$$

In practice, we use Eq. 2 in its weak form [29, 36] to check whether a given vector field $\mathcal{G}_t$ generates the target $\mu_t$:

$$\int_0^1 \int_\mathcal{H} \frac{\partial \varphi(h, t)}{\partial t} + \langle \mathcal{G}_t(h), \nabla_h \varphi(h, t) \rangle d\mu_t(h) dt = 0 \tag{3}$$

Where $\varphi \in \mathrm{Cyl}(\mathcal{H} \times [0, 1])$, and $\mathrm{Cyl}(\mathcal{H} \times [0, 1])$ is the space of smooth cylindrical test functions. Suppose that the time-varying vector field $\mathcal{G}_t$ and the induced $\mu_t$ satisfying Eq. 3 are known. We parameterize $\mathcal{G}_t$ with a neural operator $\mathcal{G}_\theta : [0, 1] \times \mathcal{H} \to \mathcal{H}$ and regress $\mathcal{G}_\theta$ to target $\mathcal{G}_t$ through flow matching objective.

$$\mathcal{L}_{\mathrm{FM}}^\dagger = \mathbb{E}_{t \sim \mathcal{U}[0,1], h \sim \mu_t} \|\mathcal{G}_\theta(t, h) - \mathcal{G}_t(h)\|^2 \tag{4}$$

However, similar to its finite-dimensional counterpart, $\mathcal{G}_t$ is typically unknown. Moreover, there are infinitely many paths of probability measures that satisfy Eq. 3 and ensure $\mu_1 \approx \nu_1$. Therefore, it is necessary to specify a path of probability measures to effectively guide the learning of $\mathcal{G}_\theta$.

By constructing the appropriate Gaussian and conditional probability measures and leveraging optimal coupling together with the dynamic Kantorovich formulation [37], we propose marginal (dynamic) optimal-transport flow matching in function space; detailed theory development and proofs are provided in Appendix F and G. In the next subsection, we show how to generalize flow matching to stochastic processes, which is induced by the function space derivation.

### 3.1 Generalizing Flow Matching to Stochastic Processes

Stochastic processes are inherently infinite-dimensional and define distributions over any collection of points (Brémaud [38], Chapter 5.1). We generalize the above marginal optimal transport flow matching on function spaces to stochastic processes by defining the transport map on any collection of points. We then show that, as the collection of points covers the space in the limit, this generalization recovers infinite-dimensional flow matching implemented with neural operators.

For any $n$ and points $\{x_1, x_2, \ldots, x_n\}$, consider an ODE system in which a vector of random variables $u_0 \in \mathbb{R}^n$ is gradually transformed into $u_1 \in \mathbb{R}^n$, for which, the $i$th entry is equal to $u(x_i)$, via a smooth, time-varying vector field, denoted by $\mathcal{G}_t$ with abuse of notation.

$$u_t := \Phi_t(u_0) = u_0 + \int_0^t \mathcal{G}_s(u_s)ds \tag{5}$$

Here, the neural operator is applied to a collection of point evaluations. Given the set of points and the density of $p_0 := p_0 (\{u_0(x_1), u_0(x_2), \ldots, u_0(x_n)\}) = \mathcal{N}(\mathbf{0}, K(\{x_1, x_2, \ldots, x_n\}))$, where $K$ is a $n \times n$ covariance matrix with entries described by kernel function $k(x_i, x_j)$ and $u_0 \sim p_0$, the time-varying density $p_t$ induced by the diffeomorphism $\Phi_t$ or $\mathcal{G}_t$ can be computed, extending the transport equation Eq. 2 to collections of points,

$$\frac{\partial p_t(u_t)}{\partial t} = -(\nabla \cdot (\mathcal{G}_t p_t))(u_t) \tag{6}$$

Eq. 6 shows that constructing $p_t$ is equivalent to constructing $\mathcal{G}_t$ for finite entries for which the analysis carries to finite collections of random variables. In the following, we refer to $p_t$ as the marginal probability path induced by $\mathcal{G}_t$ for the given collection of points. From Eq. 6, the log density can be computed through integration,

$$\log p_t(u_t) = \log p_0(u_0) - \int_0^t (\nabla \cdot \mathcal{G}_s)(u_s)ds \tag{7}$$

In this formulation, we are seeking a specific vector field that transports density $q_0$ to target density $q_1$ for any $n$ and any collection of points $\{x_1, x_2, \ldots, x_n\}$ with boundary conditions $p_0 = q_0, p_1 = q_1$. Extending optimal transport flow matching to stochastic processes, we parameterize the vector field $\mathcal{G}_t$ with a neural operator $\mathcal{G}_\theta$, which is optimized through the flow matching objective for SPL,

$$\mathcal{L}_{\text{FM}} := \sup_n \sup_{\{x_1, \ldots, x_n\}} \mathbb{E}_{t \sim \mathcal{U}(0,1), u_t \sim p_t} \|\mathcal{G}_\theta(t, u_t) - \mathcal{G}_t(u_t)\|^2 \tag{8}$$

Note that $p_t$ and $u_t$ depend on the point collocations. In the above equation, the suprema are intractable and we replace them with an expectation as a soft approximation (see Appendix A. Q5 for a detailed discussion of the approximation). Moreover, the true $\mathcal{G}_t$ is usually unknown and to address it, we derive a probability path conditioned on latent variable $z$ of the same alphabet size as the collection. Consequently, the marginal probability path $p_t(u_t)$ is a mixture of conditional probability paths $p_t(u_t|z)$,

$$p_t(u_t) = \int p_t(u_t|z)q(z)dz \tag{9}$$

$$\mathcal{G}_t(u_t) = \mathbb{E}_{q(z)}\left[\frac{\mathcal{G}_t(u_t|z)p_t(u_t|z)}{p_t(u_t)}\right]. \tag{10}$$

Given Eq. 10, the conditional flow matching (CFM) objective is defined as,

$$\mathcal{L}_{\text{CFM}} := \mathbb{E}_n \mathbb{E}_{x_1, \ldots, x_n} \mathbb{E}_{t, q(z), p_t(u_t|z)} \|\mathcal{G}_\theta(t, u_t) - \mathcal{G}_t(u_t|z)\|^2. \tag{11}$$

Eqs. 11 and 10 have an identical gradient for $\theta$, i.e. $\nabla_\theta \mathcal{L}_{\text{FM}}(\theta) = \nabla_\theta \mathcal{L}_{\text{CFM}}(\theta)$. Inspired by the finite dimensional developments [2], the variable $z$ is chosen as a couple $(u_0, u_1)$ from the coupling $\pi(u_0, u_1) = q(z)$, which is achieved by minimizing the dynamic 2-Wasserstein distance,

$$W_{\text{dyn}}(q_0, q_1)_2^2 = \inf_{p_t, \mathcal{G}_t} \int_{\mathbb{R}^n} \int_0^1 p_t(u_t)\|\mathcal{G}_t(u_t)\|^2 du_t dt \tag{12}$$

Under mild conditions on $\mathbb{R}^n$, this is equivalent to the static 2-Wasserstein distance,

$$W_{\text{sta}}(q_0, q_1)_2^2 = \inf_{\pi \in \Pi} \int_{\mathbb{R}^n \times \mathbb{R}^n} \|u_1 - u_0\|^2 d\pi(u_0, u_1). \tag{13}$$

Considering the class of Gaussian conditional probability paths $p_t(u_t|z) = \mathcal{N}(u_t|m_t(z), \sigma_t(z)^2 K(\{x_1, x_2, \ldots, x_n\}))$, with conditional flow $\Phi_t(u_0|z) = \sigma_t u_0 + m_t$. Specially, we choose $m_t = tu_1 + (1-t)u_0$ and $\sigma_t = \sigma$, where $\sigma > 0$ is a small constant. Then we have the following closed-form expression for the corresponding vector field (full derivation provided in Appendix H)

$$\mathcal{G}_t(u_t|z) = u_1 - u_0 \tag{14}$$

With the aforementioned developments, for any collection of points, we transport a Gaussian distribution to a target distribution. The Gaussian distribution is drawn from a GP, with its covariance matrix $K(x_1, \cdots, x_n)$ determined by the kernel function $k(x_i, x_j)$ of the GP. According to Kolmogorov extension theorem (KET) [39], there exists a valid stochastic process whose finite-dimensional marginal is the pushforward distribution under $\mathcal{G}_\theta$. This demonstrates that the generalization of flow matching to infinite-dimensional spaces with neural operators naturally induces the generalization of flow matching to stochastic processes. In the scenario where the limit of points covers the space, these two become equivalent. For a detailed explanation and proof, please refer to Appendix C and D.

Our framework extends seamlessly to alternative probability paths, such as those in *stochastic interpolants and rectified flow* [12, 13], because the generalization to stochastic processes is decoupled from any specific path. However, we focus on the OT path in this work for two reasons: (1) ablation and scaling studies show that it accelerates inference and enables more accurate prior learning compared to independent coupling (Table 7, and Appendix P); (2) theoretically, the OT path (straight line) simplifies the Jacobian evaluation required for likelihood estimation (discussed in the next subsection), yielding greater stability and speed than arbitrary paths.

### 3.2 Likelihood Estimation and Bayesian Universal Functional Regression

We parameterize $\mathcal{G}_\theta$ with FNO [14] to ensure our model is resolution agnostic, and assume $\mathcal{G}_\theta$ learns a map from $\nu_0$ to $\nu_1$, which serves as the prior. In practice, we deal with discretized evaluations of functions that may have different sampling rate and resolution. For instance, consider a function $u$ sampled from $\nu_1$, observed on a collection of points $u_1 := \{u(x_1), u(x_2), ..., u(x_m)\}$; thus we have a density function $\mathbb{P}(u_1)$ defined on collection of points $\{x_1, x_2, ..., x_m\}$, where $\mathbb{P}(u_1)$ is derived from measure $\nu_1$. This is similar to how a multivariate Gaussian distribution can be derived from a Gaussian measure characterized by a Gaussian process. Therefore, we can rewrite Eq. 7 as:

$$\log \mathbb{P}(u_1) = \log \mathbb{P}(u_0) - \int_0^1 (\nabla \cdot \mathcal{G}_\theta)(u_t) dt \tag{15}$$

where $u_0$ is drawn from the reference Gaussian measure $\nu_0$, which is also defined on the collection of points $\{x_1, x_2, ..., x_m\}$. Thus, $\mathbb{P}(u_0)$ is a multivariate Gaussian with a tractable density function. Furthermore, with the probability density function $\mathbb{P}(u_1)$, we can evaluate the precise likelihood of any $u_1$ from $\mathbb{P}(u_1)$ via Eq. 15. However, following a similar argument to Grathwohl et al. [40], the computation of $\nabla \cdot \mathcal{G}_\theta(u)$ incurs a cost of $\mathcal{O}(m^2)$ where $m$ is the cardinality of $\{x_1, x_2, ..., x_m\}$. This quadratic time complexity renders the likelihood calculation prohibitively expensive. To address this issue, we adopt the strategy proposed in Grathwohl et al. [40], using the unbiased Skilling-Hutchinson trace estimator [41, 42] to approximate the divergence term. This technique reduces the computation cost to $\mathcal{O}(m)$, which is the same as the cost of inference, thereby streamlining the evaluation process. The estimator is implemented as follows:

$$\nabla \cdot \mathcal{G}_\theta(u_t) = \mathbb{E}_{p(\varepsilon)}[\varepsilon^T \frac{\partial \mathcal{G}_\theta(u, t)}{\partial u} \varepsilon] \tag{16}$$

In the unbiased trace estimator, the random variable $\varepsilon$ is characterized by $\mathbb{E}(\varepsilon) = 0$ and $\text{Cov}(\varepsilon) = I$. This estimator benefits from the optimal transport nature of the map which gives rise to a direct line. The gradient computation in Eq. 16 can be efficiently handled with reverse-mode automatic differentiation, allowing for precise estimation with arbitrary error by averaging over a sufficient number of runs, which can benefit from parallel computing of GPUs.

With the efficient tool established for estimating the likelihood of any discretized function samples, we now turn our attention to UFR, i.e., Bayesian functional regression with a learned prior. Consider a collection of pointwise observations of the underlying unknown function drawn from $\nu_1$, corrupted with Gaussian noise, denoted as $\{\widehat{u}(x_1), \widehat{u}(x_2), \ldots, \widehat{u}(x_n)\}$ or $\{\widehat{u}(x_i)\}_{i=1}^n$. We specifically focus on Gaussian white noise characterized by $\epsilon \sim \mathcal{N}(0, \sigma^2)$, such that $\widehat{u}(x_i) = u(x_i) + \epsilon_i$ for $i \in \{1, \cdots, n\}$ (depending on the nature of the problem, the noise may also be considered as a correlated GP noise). In UFR setting, we are interested in the posterior distribution over new $m \geq n$ points that include the $n$ observation points.

**Proposition 3.1.** *Given noisy observations* $\{\widehat{u}(x_i)\}_{i=1}^n$, *the posterior distribution is*

$$\log \mathbb{P}\left(\{u(x_i)\}_{i=1}^m \Big| \{\widehat{u}(x_i)\}_{i=1}^n\right) = -\frac{\sum_{i=1}^n \|\widehat{u}(x_i) - u(x_i)\|^2}{2\sigma^2} + \log \mathbb{P}\left(\{u(x_i)\}_{i=1}^m\right) + C \tag{17}$$

*Where the constant $C = -\frac{n}{2} \log(2\pi\sigma^2) - \log \mathbb{P}\left(\{\widehat{u}(x_i)\}_{i=1}^n\right)$.*

*Proof.* This is derived from Bayes' theorem, along with the translation invariance property of the Gaussian distribution, see the full proof in Appendix I □

Given this form of the posterior distribution, we adopt SGLD [4] to efficiently sample from it, and then derive statistical features of interest, e.g. mean, maximum a posteriori, and posterior uncertainty, i.e., variance, from the posterior samples. More specifically, we follow the posterior sampling strategy developed by Shi et al. [1], which, given an invertible framework, suggests SGLD sampling within the input GP space where the Gaussian measure $\nu_0$ is defined and Langevin dynamics is native to, and then mapping to the data function space (where data measure $\nu_1$ is defined). We also present a scaling study in Appendix P analyzing the variance introduced by the Hutchinson trace estimator, demonstrating its robustness and effectiveness when paired with SGLD sampling. Last, Eq 17 offers a unified view of prior and posterior sampling with flow matching models by showing that when no observations are present, the posterior collapses to the prior (up to a constant), making the posterior sampling process identical to that of the prior.

Finally, we provide a plain-language summary of the framework to help the reader better understand the paper (with a detailed discussion available in Appendix Q),

- OFM is an *expressive, flow-based generative model that learns a prior over functions (stochastic process)*: a neural operator parameterizes a continuous probability flow that transports samples from a simple reference Gaussian process to data-like functions, yielding an explicit prior with a tractable density.

- It excels at functional regression by treating functions as *first-class objects* rather than mere pointwise values (unlike NPs), yielding predictions that are consistent across resolutions and arbitrary query sets.

- The learned flow is invertible, enabling change-of-variables for likelihoods and principled Bayesian regression, yielding calibrated uncertainty from few observations.

## 4  Experiments

In this section, we demonstrate the superior regression performance compared to several baselines across a variety of function datasets, including both Gaussian and highly non-Gaussian Processes. As baselines, we employ standard GP Regression [8], Deep GPs [9, 10], Conditional models [5–7], and OPFLOW [1].

For our function datasets, we analyze: (1) Gaussian and non-Gaussian with known posterior, including 1D GPs, 2D GPs, and 1D Truncated GPs, (TGP). (2) Highly non-GPs, datasets with unknown posterior, such as those derived from Navier-Stokes equations [14], black hole dataset from expensive Monte Carlo simulation, and 2D Signed Distance Functions extracted from MNIST digits (MNIST-SDF) [43]. During regression, we assume that the prior $\mathcal{G}_\theta$ is always successfully trained and remains frozen. Details about the learning process for priors and experimental setup for regression are provided in the Appendix M,  O.

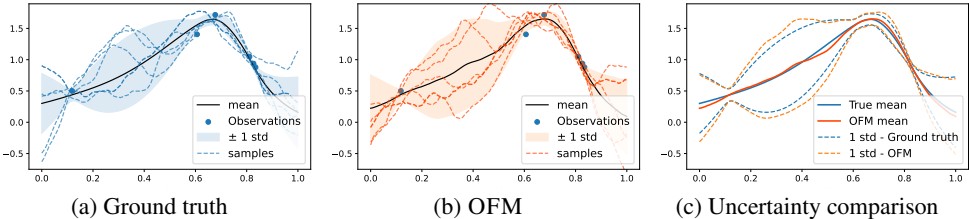

| (a) Ground truth | (b) OFM | (c) Uncertainty comparison |

Figure 3: OFM regression on GP data. (a) Ground truth GP regression with observed data and predicted samples. (b) OFM regression with observed data and predicted samples. (c) Standard deviation comparison between true GP and OFM predictions.

**1D GP data.** This experiment replicates the results of classical GP regression, wherein the posterior distributions are precisely known in a closed form. The process involves generating a single new

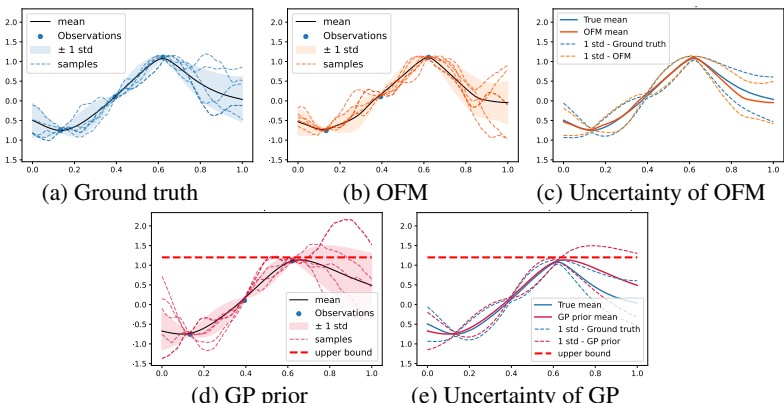

(a) Ground truth        (b) OFM        (c) Uncertainty of OFM

(d) GP prior        (e) Uncertainty of GP

Figure 4: OFM regression on TGP data.

realization from the data measure $\nu_1$. We then select observations at $n = 6$ randomly chosen positions, incorporating a predefined noise level. The posterior is inferred across $m = 128$ positions, which includes estimating noise-free values at the observation points. We evaluate our results with two commonly used quantities in the GP literature (1) Standardized Mean Squared Error (SMSE) that normalizes the mean squared error by the variance of the ground truth; and (2) Mean Standardized Log Loss (MSLL), originally introduced by Williams and Rasmussen [8], defined as:

$$-\log p(\{u(x_i)\}_{i=1}^m | \{\widehat{u}(x_i)\}_{i=1}^n) = \frac{1}{2}\log(2\pi\sigma^2) + \frac{(\{u(x_i)\}_{i=1}^m - \{\bar{u}(x_i)\}_{i=1}^m)^2}{2\sigma^2} \qquad (18)$$

where $\{\widehat{u}(x_i)\}_{i=1}^n$ represents observations, $\{x_i\}_{i=1}^m$, $\{u(x_i)\}_{i=1}^m$, indicate the new positions queried, and the test data (true posterior samples). Meanwhile, $\{\bar{u}(x_i)\}_{i=1}^m$ and $\sigma^2$ are predicted mean and variances from the model. We average out SMSE and MSLL over a test dataset containing 1000 true GP posterior samples for all models. The performance of each model is detailed in Table 1. From Fig. 3, the regression with OFM matches the analytical solution well and provides realistic posterior samples. We further include GP regression tasks using more complex kernels (Gibbs and Rational Quadratic kernel), as shown in Appendix N, OFM consistently outperforms all comparative methods across all metrics.

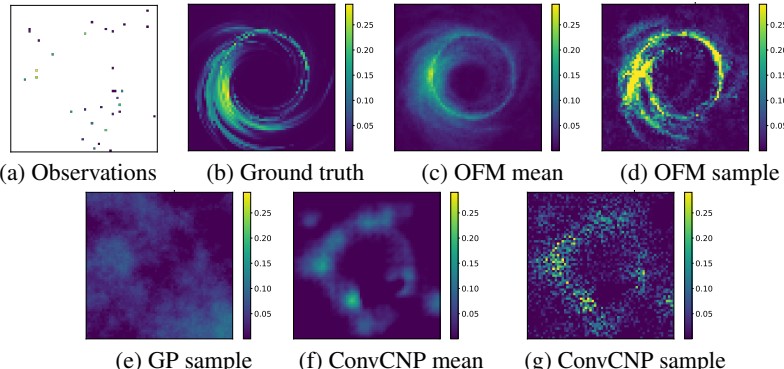

(a) Observations    (b) Ground truth    (c) OFM mean    (d) OFM sample

(e) GP sample    (f) ConvCNP mean    (g) ConvCNP sample

Figure 5: OFM regression on black hole data with resolution $64 \times 64$. (a) 32 random observations. (b) Ground truth sample. (c) Predicted mean from OFM. (d) One posterior sample from OFM. (e) One posterior sample from best fitted GP. (f) Predicted mean from ConvCNP. (g) One posterior sample from ConvCNP.

**Truncated GP data.** In this experiment, we analyze the regression performance of OFM for tractable non-GP. Specifically, we work on truncated GP [1, 44], which constrains the function amplitude within a specified range. This is achieved by applying a sampling-rejection strategy on samples from the GP prior. We set the bounds of our TGP to $[-1.2, 1.2]$ and perform regression using observations only at three points, while estimating the posterior across $m = 128$ points. Subsequently, we sample 1000 true TGP posteriors from the GP prior to calculate the mean and standard deviation. Traditional metrics like MSLL and SMSE, which assume a Gaussian posterior, are not suitable for TGP. We

Table 1: Comparison of OFM with baseline models: GP regression; OpFlow [1]; Conditional models: NP ( [7]); Attentive NP ( [5], ANP); Convolutional Conditional NP ( [6], ConvCNP); Deep variational GP ( [9], DGP); Deep Sigma Point Process ( [10], DSSP); Metrics SMSE and MSLL used for 1D and 2D GP example. Mean squared error for the predicted mean ($\mu$) and standard deviation ($\sigma$) are used for TGP example. Performance of GP regression for 1D and 2D GP are removed (marked with '$-$'), which are taken as the ground truth. Best performance in bold.

| Dataset $\rightarrow$ | 1D GP | | 2D GP | | 1D TGP | |
|---|---|---|---|---|---|---|
| Algorithm $\downarrow$ Metric $\rightarrow$ | SMSE | MSLL | SMSE | MSLL | $\mu$ | $\sigma$ |
| GP prior | - | - | - | - | $6.4 \cdot 10^{-2}$ | $1.6 \cdot 10^{-2}$ |
| NP | $6.1 \cdot 10^{-1}$ | $4.5 \cdot 10^{0}$ | $1.7 \cdot 10^{-1}$ | $2.1 \cdot 10^{0}$ | $1.0 \cdot 10^{-1}$ | $1.9 \cdot 10^{-2}$ |
| ANP | $5.1 \cdot 10^{-1}$ | $9.8 \cdot 10^{-1}$ | $1.6 \cdot 10^{-1}$ | $1.1 \cdot 10^{0}$ | $1.4 \cdot 10^{-1}$ | $1.7 \cdot 10^{-2}$ |
| ConvCNP | $5.6 \cdot 10^{-1}$ | $2.7 \cdot 10^{-1}$ | $1.7 \cdot 10^{-1}$ | $4.5 \cdot 10^{-1}$ | $1.6 \cdot 10^{-2}$ | $2.1 \cdot 10^{-3}$ |
| DGP | $4.1 \cdot 10^{-1}$ | $6.8 \cdot 10^{-2}$ | $1.8 \cdot 10^{0}$ | $4.2 \cdot 10^{0}$ | $4.9 \cdot 10^{-1}$ | $1.4 \cdot 10^{-2}$ |
| DSPP | $4.7 \cdot 10^{-1}$ | $6.5 \cdot 10^{0}$ | $1.9 \cdot 10^{-1}$ | $6.6 \cdot 10^{0}$ | $1.1 \cdot 10^{-2}$ | $1.3 \cdot 10^{-2}$ |
| OpFlow | $5.0 \cdot 10^{-1}$ | $2.0 \cdot 10^{-1}$ | $1.4 \cdot 10^{-1}$ | $\mathbf{1.1 \cdot 10^{-1}}$ | $1.3 \cdot 10^{-2}$ | $3.9 \cdot 10^{-3}$ |
| **OFM(Ours)** | $\mathbf{4.1 \cdot 10^{-1}}$ | $\mathbf{5.5 \cdot 10^{-2}}$ | $\mathbf{1.3 \cdot 10^{-1}}$ | $1.6 \cdot 10^{-1}$ | $\mathbf{5.2 \cdot 10^{-3}}$ | $\mathbf{9.5 \cdot 10^{-4}}$ |

evaluate the performance using the mean squared error for both the predicted mean and standard deviation. The results are reported in Table. 1, and illustrated in Fig. 4. OFM accurately learns the specified bounds and provides accurate estimations of mean and standard deviation, along with realistic posterior samples. In contrast, directly applying GP regression exceeds the bounds and yields unrealistic posterior samples.

**2D GP data.** Similar to the 1D GP example, we extend our regression analysis to 2D GP data. As shown in Fig. 6 and detailed in Table 1, OFM provide accurate posterior estimation. The relative error shown in Fig. 6 is the absolute error normalized by the maximum absolute value of the mean prediction derived from the ground truth GP regression.

**Navier-Stokes, Black hole and MNIST-SDF datasets.** We collected a 2D Navier-Stokes dataset and applied OFM for the regression. Unlike the GP experiments, where MSLL and SMSE score serve as standard benchmarks, evaluating the performance of models on general non-GPs presents a significant challenge due to the difficulty of determining the true posterior and lack of benchmarks. Moreover, the evidence term in the posterior (Eq. 41) is intractable, so the likelihood cannot serve as a meaningful evaluation metric in our setting. A detailed discussion is provided in Appendix A.

We present the predicted mean and a posterior sample in Fig 2 for visual comparison with the ground truth. The predicted mean and the posterior sample are closely aligned with the ground truth. In contrast, traditional GP regression and NP models failed to accurately capture the dynamics of the Navier-Stokes data. In Fig. 5, we conduct a similar analysis using a simulated black hole dataset. Here, OFM provides a much more realistic mean and posterior sample that capture the density and swirling patterns of the black hole. Once again, GP regression and NP fail to capture these key statistics. We observe similar outcomes when applying OFM to the MNIST-SDF example (Fig 8), where OFM correctly recognizes the number "7" while GP regression does not.

# 5   Conclusion

In this paper, we proposed Operator Flow Matching (OFM) for stochastic process learning, which generalizes flow matching models to infinite-dimensional space and stochastic process with optimal transport path. OFM efficiently computes the probability density for any finite collection of points and supports mathematically tractable functional regression. We extensively tested OFM across a diverse range of datasets, including those with closed-form GP and non-GP data, as well as highly non-GP such as Navier-Stokes and black hole data. In comparative evaluations, OFM consistently outperformed all baseline models, establishing new standards in stochastic process learning and regression.

Despite SOTA accuracy, our method is presently limited to low-dimensional domains and demands larger datasets and more compute than GP-based baselines. see Appendix A, O and Q for details. Python code available at `https://github.com/yzshi5/SPL_OFM`

**Acknowledgments**

This material is based upon work supported by the U.S. Department of Energy, Office of Science, Office of Advanced Scientific Computing Research, Science Foundations for Energy Earthshot under Award Number DE-SC0024705. ZER is supported by a fellowship from the David and Lucile Packard Foundation. We also thank Charles Gammie, Ben Prather, Abhishek Joshi, Vedant Dhruv, and Chi-kwan Chan for providing the black hole simulations.

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

# A  Potential questions and answers

Our method combines many vital fields, including

- **Operator learning**
- **Flow-based generative models (flow matching)**; **Optimal transport in function space**
- **Bayesian uncertainty**; **Gradient Markov chain Monte Carlo, SGLD**; **Stochastic trace estimation**
- **Gaussian processes**; **Stochastic processes (Kolmogorov extension theorem)**

Because readers will have diverse backgrounds and expertise, we provide some potential questions and answers to enhance the clarity and highlight our contributions.

**Q1: Why is it hard to obtain the true prior and posterior of an unknown stochastic process, and what would we gain if we could?**

*A1:* Generalizing GP regression to arbitrary stochastic process regression has remained a long-standing challenge that has kept many researchers busy for decades. The core difficulties are twofold: first, learning a general stochastic-process prior from historical data with an expressive enough model, and second, deriving both the exact posterior distribution and an efficient sampling scheme.

Prior work falls into three main categories—GPs, deep GPs, and conditional models (NP family). Only the classical GP can fully characterize a stochastic process, and then only when the exact GP prior (mean function, covariance kernel, all hyper-parameters, and observation-noise variance) is already known. Even fitting one GP to data generated by another GP with unknown parameters rarely yields the true posterior. Deep GPs and NPs possess stronger representational power compared to GP, but **representational power $\neq$ exactness**: their optimization rely on approximate posterior and **DO NOT** produce a closed-form "true posterior."

If the true prior and posterior were available, we would gain near-complete control over general stochastic process, enabling perfect uncertainty quantification and Bayes-optimal decisions, which has numerous applications in finance market, science and engineering problems

**Q2: How does OFM differ from standard finite-dimensional flow matching? What does it mean to generalize flow matching to stochastic processes, and how does extending a generative model to a function space differ from extending it to a full stochastic process?**

*A2:* A standard flow matching model learns a transport map between two distributions defined on a fixed grid (e.g., a pixel lattice). Consequently, it can only generate samples at that specific resolution. In contrast, OFM learns a map between two stochastic processes in function space. This endows it with several properties that standard flow matching lacks:

 (i)  **Resolution agnosticism**. OFM learns a transport map between two distributions defined on any given collection of points, without regard for the number of points, or their locations in the domain. This enables capabilities like zero-shot generation without retraining.
 (ii) **Stochastic process consistency**. OFM respects the metric of the underlying space, ensuring that points close to each other in the input domain have appropriately correlated values in the output distribution. This is part of learning a valid stochastic process, which also satisfies crucial theoretical properties like the Kolmogorov extension theorem (i.e, consistency under marginalization)
(iii) **Convergence to a continuous function**. As the collection of query points becomes denser within the domain, the output of OFM converges to the underlying continuous function.
(iv) **Backwards compatibility**. When evaluated on a fixed set of points, OFM recovers the behavior of a conventional (finite-dimensional) flow matching up to a linear transformation.

Extending a generative model (e.g., flow matching) to an *infinite-dimensional function space* and extending it to a *full stochastic process* are related but distinct goals. The first concerns generating function values at any finite collection of points; Lebesgue measure and Kolmogorov consistency come into play, but the focus remains on finite dimensional marginals. The second addresses the

distribution of the entire function itself, requiring a probability measure over an infinite-dimensional space.

Because of this distinction, lifting a model merely to a *function space* is strictly weaker. For example, GANO [22] extends GANs to function space but not to a stochastic process, as it cannot describe the joint law of function values at arbitrary query sets. In OFM, we bridge these two viewpoints for flow matching with neural operators: by exploiting the discretization-convergence property of neural operators and the invertibility of flow matching, we extend the method to both function spaces and stochastic processes. See Appendix C and D for details.

**Q3: These methods may appear similar to image inpainting or restoration with flow- or diffusion-based models; could you elaborate on the differences?**

*A3:* Current inpainting with flow/diffusion models treats the task as an inverse problem in a finite-dimensional setting: one learns to reconstruct missing pixels on a fixed resolution grid, effectively solving a regression problem at a predetermined set of points. By contrast, OFM operates directly on *functions* and is *resolution-agnostic*. Given the same partial observations, our model can predict the entire function at any collection of query points, coarse or fine.

Moreover, OFM delivers principled uncertainty: its posterior quantifies the full distribution of possible completions, whereas finite-dimensional inpainting methods typically rely on an ensemble of visually plausible samples whose variability is not a calibrated measure of uncertainty. Finally, OFM remains effective even when observations are extremely sparse. (e.g $0.7\%$ of total observations) —a regime in which grid-based inpainting approaches struggle.

**Q4 : Could you elaborate further on the connections to related work? The approach appears to intersect with several studies, including OPFLOW [1], COT-FM [45], OT-CFM [2], and others.**

*A4:* Because our work combines multiple fields, it naturally has connections with many other studies. Readers are referred to the appendix Q for a detailed discussion.

**Q5: In Eq 11. What is the argument of replacing the superma in Eq 8 with expecations? When is this valid?**

*A5:* The supremum in Eq. 8 represents a worst-case error over the entire function space, which is computationally intractable to optimize directly. We therefore relax this hard constraint by replacing the supremum with an expectation, as formulated in Eq 11. This is a common empirical consideration.

Instead of minimizing the worst-case error, our objective becomes minimizing the tractable average-case error across the distribution. Minimizing the error on average provides a strong practical incentive for the model to perform well across the entire function space, thereby effectively reducing the worst-case error. Such replacement is always valid under the weaker goal. The validity of this approach as a tractable proxy is further confirmed by our empirical results, which show it successfully guides the model to learn the intended functional mapping (Appendix M).

**Q6: Why not use log-likelihood as the evaluation metric for non-GP regression tasks when comparing to NP models? And why not include more recent conditional model baselines, such as NDP [34]**

*A6:* There are several reasons that likelihood as an evaluation metric is not relevant in our case. First, as shown in Eq 41, there is an evidence term, which is a constant, in the posterior distribution in OFM framework. The constant evidence term is intractable but does not contribute to MAP estimation, mean estimation, and posterior sample in general.

Second, even if we can compute the evidence term, we still cannot make the comparison. The graphical model in the NP is such that the conditional model is trained using the MLE principle and the learned likelihood model dependent. The posterior in OFM provides the posterior using the Bayes rule, utilizes a different model, and has a different graphical model. These quantities are not directly relevant to be compared.

Third, computing the true posterior for a general stochastic process is a known challenging problem. Due to the complexity nature of the problem, there doesn't exist a well-recognized metric. The

evaluation of quality of posterior performance requires domain knowledge from experts and varies case by case, which is an exciting research direction.

For baseline models,We already include a broad set of SOTA baselines. On the operator-learning side, the latest OPFLOW [1] (2024) is covered. For deep GPs, we adopt Doubly Stochastic Variational Deep GPs [9] and the Deep Sigma-Point Process [10], both widely recognized and backed by publicly reproducible code.

Within the NP family, ConvCNP [6] remains a standard benchmark, and we include two additional NP variants. Our baselines are limited to models with publicly reproducible implementations. Although we attempted to add the newer NDP [34], we encountered significant reproducibility issues with the authors' code—an obstacle reported by others as well. Instead, we offer a detailed theoretical comparison between OFM and NDP in Appendix Q.

**Q7: What do you mean by an "exact posterior" and a practical solution? How can I apply the model to my own tasks, and what do I need to prepare?**

*A7:* For functional regression, posterior error has two sources for any method: **(1) formulation error**—the gap between the model's theoretical posterior and the true one—and **(2) approximation error** from finite data, limited capacity, and optimization. Deep GPs rely on variational inference, optimizing an ELBO and thereby introducing a formulation gap: the posterior is only an *approximation*. NPs make even stronger simplifying assumptions, and doesn't provide the true posterior. All models, including OFM, incur approximation error, but the discussion above concerns only **formulation error**.

A *practical* solution means an expressive backbone that can learn a complex stochastic-process prior and a posterior-sampling routine whose runtime and memory footprint remain acceptable for typical users; see Appendix O for quantitative details.

Using the model is straightforward. OFM offers GP-style regression for non-GP tasks: given noisy observations, it returns the posterior at arbitrary query points. The key difference is that a GP prior is fixed by a hand-tuned kernel, whereas OFM learns the stochastic-process prior directly from data. For details on prior learning and SGLD-based posterior sampling, see Appendices M and L.

## B  Background: Flow Matching, Gaussian Measures on Function Spaces, and the Cameron–Martin Theorem

In this section, we provide essential background and high-level intuitive explanations of the topics involved in this paper for readers.

**Flow Matching.** Flow matching is a state-of-the-art generative paradigm that learns a time-dependent velocity field whose ODE transports samples from a simple reference (e.g., Gaussian) to the target data distribution, yielding an unbiased, simulation-free training objective that directly regresses ground-truth velocities along probability paths.

The approach is tightly linked to physics via the continuity equation, and—when cast as a continuous normalizing flow—provides a (deterministically) invertible transformation with tractable likelihoods, while also admitting stochastic variants when desired. In practice it matches diffusion-level quality with faster training and sampling (often few- or even single-step generation) and excellent scalability, which is why it has been adopted in challenging domains such as large-scale video generation, in-context image generation, and protein ensemble generation [46, 47]. We therefore use flow matching as the foundation for prior learning, leveraging its simple training objective, physics-grounded structure, and strong empirical performance.

**Gaussian measures.** A probability measure $\mu$ on a separable Hilbert space $\mathcal{H}$ is *Gaussian* if for any finite collection of vectors $\{h_1, \ldots, h_n\} \subset \mathcal{H}$, the random vector $(\langle X, h_1 \rangle, \ldots, \langle X, h_n \rangle)$ has a multivariate Gaussian distribution, where $X \sim \mu$. The family $\{\langle X, h \rangle : h \in \mathcal{H}\}$ is therefore a *Gaussian process* indexed by $\mathcal{H}$, with mean $h \mapsto \langle m, h \rangle$ for some $m \in \mathcal{H}$. Conversely, if a Gaussian process $\{X(t) : t \in T\}$ has sample paths that almost surely belong to a function space $\mathcal{H}$ (e.g., $C(T)$ or $L^2(T)$), then the law of the random path $t \mapsto X(t)$ is a *Gaussian measure* on $\mathcal{H}$. In short: a Gaussian measure is the path-space law of a GP, and a GP is the collection of linear probes of a Gaussian measure.

**Cameron–Martin space and theorem.** Let $\mu$ be a Gaussian measure on a separable Hilbert space $\mathcal{H}$ with mean $m \in \mathcal{H}$ and covariance operator $C : \mathcal{H} \to \mathcal{H}$ (self-adjoint, positive, trace-class). The *Cameron–Martin space* (the RKHS associated with $\mu$) is

$$\mathcal{H}_\mu = \overline{\mathrm{Range}(C^{1/2})} \subseteq \mathcal{H},$$

equipped with the inner product $\langle u, v \rangle_{\mathcal{H}_\mu} := \langle C^{-1/2}u, C^{-1/2}v \rangle_{\mathcal{H}}$ on $\mathrm{Range}(C^{1/2})$, extended by completion. With this choice, the inclusion $\mathcal{H}_\mu \hookrightarrow \mathcal{H}$ is continuous. For $h \in \mathcal{H}$, write $\mu_h(A) := \mu(A - h)$ for the translate. The *Cameron–Martin theorem* states:

(i) If $h \in \mathcal{H}_\mu$, then $\mu_h$ is absolutely continuous with respect to $\mu$ (in fact, $\mu_h$ and $\mu$ are equivalent), with

$$\frac{d\mu_h}{d\mu}(x) = \exp\Big( \langle C^{-1/2}h, C^{-1/2}(x - m) \rangle_{\mathcal{H}} - \tfrac{1}{2} \|h\|_{\mathcal{H}_\mu}^2 \Big) \quad \text{for } \mu\text{-a.e. } x,$$

where the pairing is understood in the Paley–Wiener sense (which in finite dimensions reduces to $\langle h, C^{-1}(x - m) \rangle$).

(ii) If $h \notin \mathcal{H}_\mu$, then $\mu_h \perp \mu$ (mutually singular).

In finite dimensions this reduces to the classical mean-shift formula for $\mathcal{N}(m, \Sigma)$ with $C = \Sigma$, and $\|h\|_{\mathcal{H}_\mu}^2 = \langle h, \Sigma^{-1}h \rangle$. Thus, $\mathcal{H}_\mu$ pinpoints exactly the directions along which a Gaussian measure can be translated while remaining (mutually) absolutely continuous.

# C   Stochastic process learning

Let $(\Omega, \mathcal{F}, P)$ denote a probability space and let $(\mathbb{R}^d, \mathcal{B}(\mathbb{R}^d))$ denote a measurable space where $\mathcal{B}(\mathbb{R})$ is the Borel space. Following the standard definition of stochastic processes (Brémaud [38], Chapter 5.1), a stochastic process $\mathcal{P}$ on a domain $D$ is a collection of $\mathbb{R}^d$-valued random variables indexed by members of $D$, i.e.,

$$\{a(x) : x \in D\}$$

jointly following the probability law $P$. In the special case of Gaussian processes, e.g., Wiener process, following the Gaussian law for $P$, for any collection points $\{x_1, x_2, \ldots, x_n\}$, the random variables $\{a(x_1), a(x_2), \ldots, a(x_n)\}$ are jointly Gaussian, resulting in a function $a$ to be drawn from a GP. We need to emphasize, $\{a(x_1), a(x_2), \ldots, a(x_n)\}$ is a collection of random variables (random vector) equipped with Lebesgue measure, and represents a discretized observation of one continuous function $a$. In practice, the joint probability distribution of the collection of the random variables is unknown a priori, and needs to be learned.

In SPL, one way we suggest is to learn an invertible operator $\mathcal{T}$ that maps a base stochastic process $\mathcal{P}$ to another stochastic process $\mathcal{Q}$ that represents the data via discretization convergence theorem (see Appendix D). That is, for any collection of points $\{x_1, x_2, \ldots, x_n\}$, and for any $n$, the operator $\mathcal{T}$ maps the law on $\{a(x_1), a(x_2), \ldots, a(x_n)\}$ to $\{u(x_1), u(x_2), \ldots, u(x_n)\}$ and vice versa for the inverse $\mathcal{T}^{-1}$, where $u(x)$ is a pointwise evaluation of function data sample, i.e.,

$$\{u(x_1), u(x_2), \ldots, u(x_n)\} = \mathcal{T}\left(\{a(x_1), a(x_2), \ldots, a(x_n)\}\right)$$

Then, the probability of $\{u(x_1), u(x_2), \ldots, u(x_n)\}$, at evaluation points $\{x_1, x_2, \ldots, x_n\}$, for any $n$ and collection of points on $D$ is given by,

$$\mathbb{P}\left(\{u(x_1), u(x_2), \ldots, u(x_n)\}\right) = \mathbf{J}\mathcal{T}\Big|_{\{a(x_1), a(x_2), \ldots, a(x_n)\}} \mathbb{P}\left(\{a(x_1), a(x_2), \ldots, a(x_n)\}\right)$$

where with abuse of notation $\mathbb{P}(u(x))$ denotes the density of $u(x)$ at point $x$, same for $\mathbb{P}(a(x))$, and similarly, following the notation in Theorem 11.1 of Villani [48], $\mathbf{J}\mathcal{T}\Big|_{\{a(x_1), a(x_2), \ldots, a(x_n)\}}$ is the absolute value of the Jacobian determinant of the map from the random vector $\{a(x_1), a(x_2), \ldots, a(x_n)\}$ at points $\{x_1, x_2, \ldots, x_n\}$ to the random vector $\{u(x_1), u(x_2), \ldots, u(x_n)\}$ via inverse operator $\mathcal{T}^{-1}$. We further show that the pushedforward $\mathcal{Q}$ is indeed a valid stochastic process via Kolmogorov Extension Theorem (KET) [39] with a proof provided in Appendix. D . In SPL, we aim to learn a neural operator $\mathcal{T}_\theta$ such that the resulting $\mathcal{Q}$ matches the data process under the true $\mathcal{T}$.

# D  Model stochastic process with infinite-dimensional flow matching via Kolmogorov Extension Theorem

In operator learning, neural operators [3, 14, 15] are typically designed to map an input function to an output function. When the input function is provided at a specific discretization (e.g., a set of points with their corresponding values), the model processes this discretized input as a collection of points and their values. Traditionally, in operator learning, this process is seen as an approximation of the operator's application to the underlying continuous function, where the discretization introduces approximation errors. Thus, the input is conceptually still treated as a function.

Moreover, the application of the operator to a collection of points is well-defined, and, by the discretization convergence theorem, as the number of points increases, this operation converges to a well-defined mapping. In this paper, leveraging these properties, we adopt a different perspective as described in the introduction. We extend neural operators to define explicit maps between collections of points. In this framework, the input is not the abstract function itself but rather a collection of points and their associated values. Importantly, this mapping remains well-defined regardless of the number of points in the collection and, by the discretization convergence theorem, converges to a unique mapping as the point collection approaches the underlying continuous function.

Next, we show that given an invertible operator $\mathcal{T}$ and a valid stochastic process $\mathcal{P}$ whose finite dimensional marginal is $\mathbb{P}(\{a(x_1), a(x_2), ..., a(x_n)\})$, there exist a valid stochastic process $\mathcal{Q}$ with finite-dimensional marginal $\{u(x_1), u(x_2), \ldots, u(x_n)\}$.

Once again, as defined in Section C

$$\{u(x_1), u(x_2), \ldots, u(x_n)\} = \mathcal{T}\left(\{a(x_1), a(x_2), \ldots, a(x_n)\}\right)$$

Then, the probability of $\{u(x_1), u(x_2), \ldots, u(x_n)\}$, at evaluation points $\{x_1, x_2, \ldots, x_n\}$, for any $n$ and collection of points on $D$ is given by,

$$\mathbb{P}\left(\{u(x_1), u(x_2), \ldots, u(x_n)\}\right) = \mathbf{J}\mathcal{T}\Big|_{\{a(x_1), a(x_2), \ldots, a(x_n)\}} \mathbb{P}\left(\{a(x_1), a(x_2), \ldots, a(x_n)\}\right) \quad (19)$$

where with abuse of notation $\mathbb{P}(u(x))$ denotes the density of $u(x)$ at point $x$, same for $\mathbb{P}(a(x))$, and similarly, following the notation in Theorem 11.1 of Villani [48], $\mathbf{J}\mathcal{T}\Big|_{\{a(x_1), a(x_2), \ldots, a(x_n)\}}$ is the absolute value of the Jacobian determinant of the map from the random vector Jacobian of the map from the random vector $\{a(x_1), a(x_2), \ldots, a(x_n)\}$ at points $\{x_1, x_2, \ldots, x_n\}$ to the random vector $\{u(x_1), u(x_2), \ldots, u(x_n)\}$ via inverse operator $\mathcal{T}^{-1}$. We should notice Eq. 19 represents the changes of variables between two random vectors, with Lebesgue measure involved. The connection between the finite-dimensional marginal (equipped with Lebesgue measure) and the probability measure of a stochastic process in infinite-dimensional space is described by Kolmogorov Extension theorem (KET) [39], which assures that if all finite-dimensional distributions (i.e., distributions of function at finite collection of points) are consistent, then a stochastic process exists that matches finite-dimensional distributions.

Formally, according to KET, to establish that a valid stochastic process $\mathcal{Q}$, which has $\mathbb{P}\left(\{u(x_1), u(x_2), \ldots, u(x_n)\}\right)$ as its finite dimensional distributions, it is essential to demonstrate that such a joint distribution satisfies the following two consistency properties:

**Permutation invariance.** For any permutation $\pi$ of $\{1, \cdots, n\}$, the joint distribution should remain invariant when elements of $\{x_1, \cdots, x_n\}$ are permuted, such that

$$\mathbb{P}\left(\{u(x_1), u(x_2), \ldots, u(x_n)\}\right) = \mathbb{P}\left(\{u(x_{\pi(1)}), u(x_{\pi(2)}), \ldots, u(x_{\pi(n)})\}\right) \quad (20)$$

**Marginal Consistency.** This principle specifies that that if a portion of the set is marginalized, the marginal distribution will still align with the distribution defined on the original set, such that for $m \geq n$

$$\mathbb{P}\left(\{u(x_1), u(x_2), \ldots, u(x_n)\}\right) = \int \mathbb{P}\left(\{u(x_1), u(x_2), \ldots, u(x_m)\}\right) du(x_{n+1}) \cdots du(x_m) \quad (21)$$

The permutation invariance property is naturally upheld when utilizing operator, as there is no inherent order among the elements in the set $\{x_1, x_2, \ldots, x_n\}$. Furthermore, the marginal

consistency property is also maintained due to the definition of operator $\mathcal{T}$ (see Eq. 19), which ensures that $\mathbb{P}\left(\{u(x_1), u(x_2), \ldots, u(x_n)\}\right)$ is closed under marginalization. This is because $\mathbb{P}\left(\{a(x_1), a(x_2), \ldots, a(x_n)\}\right)$ is closed under marginalization, which fully determines $\mathbb{P}\left(\{u(x_1), u(x_2), \ldots, u(x_n)\}\right)$ through the Jacobian. While verifying that $\mathcal{Q}$ constitutes a valid induced stochastic process is straightforward given the $\mathcal{T}$, approximating the $\mathcal{T}$ with a neural operator is non-trivial and depends highly on the model used (related to expressiveness). For instance, in Transforming GP [33], the authors employ a marginal normalizing flow, which acts as a point-wise operator to transform values from a GP to another. Consequently, the induced Jacobian is a diagonal matrix. More recently, OpFlow [1] introduces an invertible neural operator by generalizing RealNVP to function space, which induces a triangular Jacobian matrix. In our work, we extend this framework to a more comprehensive case: a diffeomorphism. Here, the induced Jacobian is a full-rank matrix and is not necessarily triangular or diagonal, the determinant of the Jacobian for any collection of points is calculated through Eq 15.

Last, we want to clarify the the connection between the notions of operator $\mathcal{T}$ and operator $\mathcal{G}$ throughout this paper. The operator $\mathcal{T}$ is the $\Phi_t$ (a diffemporhism) defined in Eq. 5, which is the integral of $\mathcal{G}$ over time interval $[0, 1]$. Due to the nature of an ODE system, $\mathcal{T}$ is invertible. However, $\mathcal{G}$ is not necessary invertible, which enables us to parameterize it with a classical neural operator, like FNO [14].

# E   Universal Functional Regression

UFR is concerned with Bayesian regression on function spaces [1], where it can be used to infer the posterior of an unknown function on a domain $D$ from a collection of pointwise observations. The observations are often corrupted with noise of variance $\sigma^2$, denoted as $\{\widehat{u}(x_1), \widehat{u}(x_2), \ldots, \widehat{u}(x_n)\}$ or $\{\widehat{u}(x_i)\}_{i=1}^{n}$. More specifically, for $m \geq n$ points at which the function is to be inferred,

$$\mathbb{P}\left(\{u(x_1), u(x_2), \ldots, u(x_m)\} \Big| \{\widehat{u}(x_1), \widehat{u}(x_2), \ldots, \widehat{u}(x_n)\}\right)$$

Note that when the prior over the function space is Gaussian, UFR reduces to the celebrated GP regression. Following Bayes rule, and maps between stochastic processes, we obtain the log posterior as follows,

$$\log \mathbb{P}\left(\{u(x_i)\}_{i=1}^{m} \Big| \{\widehat{u}(x_i)\}_{i=1}^{n}\right) = -\frac{1}{2}\sum_{i}^{n} \frac{(\widehat{u}(x_i) - u(x_i))^2}{\sigma^2} - n\log(\sigma) - \frac{n}{2}\log(2\pi)$$
$$+ \log \mathbb{P}\left(\{u(x_i)\}_{i=1}^{m}\right) - \log \mathbb{P}\left(\{\widehat{u}(x_i)\}_{i=1}^{n}\right)$$

This equality holds for any collection of points. It is worth noting that the posterior is exact up to constants, i.e., the second, third, and last terms are constant. Therefore, they do not contribute in MAP estimation, mean estimation, and functional regression in general, and there is no need to compute them.

# F   Marginal (dynamic) optimal-transport flow matching in function space via optimal coupling and dynamic Kantorovich formulation

Consider a joint probability measure $\pi(\nu_0, \nu_1)$ on $\mathcal{H} \times \mathcal{H}$, where the reference measure $\nu_0$, is chosen as a Gaussian measure, whose absolute continuity is well-studied [49]. We characterize $\nu_0$ by a GP with trace-class covariance operator. e.g. $\nu_0 = \mathcal{N}(m_0, C_0)$, where $m_0$ is the mean, $C_0$ is the covariance operator. With the joint measure $\pi(\nu_0, \nu_1)$, we sample a function pair $z := (h_0, h_1)$.

Assuming $\nu_1$ has full support on the Cameron-Martin space associated with $\nu_0$ (following the convention of literatures [26, 29]), we construct a conditional probability measure $\mu_t(\cdot|z)$ as a Gaussian measure with trace-class covariance operator and small operator norm to approximate Dirac measures in the sense of weak convergence. Such that, at $t = 0$ and $t = 1$, $\mu_t(\cdot|z)$ is a centered around $h_0, h_1$, approximating $\delta_{h_0}, \delta_{h_1}$ respectively; Subsequently, we can construct a new marginal probability measure by mixing these approximated Dirac measures:

$$\mu_t(A) = \int \mu_t(A|z)d\pi(z), \ \forall A \in \mathcal{B}(\mathcal{H}) \tag{22}$$

Due to $d\pi(z)$ being always positive, the conditional probability measure (Dirac measure approximated by Gaussian measure) is absolutely continuous with respect to $\mu_t$. Eq. 22 indicates that $\mu_0 = \int \delta_{h_0} d\pi(z) \approx \nu_0$, and $\mu_1 = \int \delta_{h_1} d\pi(z) \approx \nu_1$. This formulation suggests that $\mu_0, \mu_1$ represent convolutions of $\nu_0, \nu_1$ with Gaussian measures. For a more detailed discussion on convolution with Gaussian measures, we refer the readers to Appendix B.1 of Lim et al. [26].

Please note, dirac measure is not a necessary condition for Eq. 22; the only constraint is the boundary conditions. One viable choice for the conditional measure is a Gaussian measure with a small operator norm, which (in fact) approximates the Dirac measure in the weak convergence sense. Alternative probability path, as described in Lipman et al. [11], Albergo and Vanden-Eijnden [12], Liu et al. [13] are equally valid.

Suppose $\int_0^1 \int_{\mathcal{H}} \int_{\mathcal{H} \times \mathcal{H}} \|\mathcal{G}_t(h|z)\| d\mu_t(h|z) d\pi(z) dt$ is finite to guarantee the vector field is sufficiently regular (Lipschitz continuity), where $\mathcal{G}_t(\cdot|z)$ is the conditional vector field. Under this condition, the vector field that generates $\mu_t$ as specified in Eq. 22 and Eq. 3 can be expanded as follows :

$$\mathcal{G}_t(h) = \int_{\mathcal{H} \times \mathcal{H}} \mathcal{G}_t(h|z) \frac{d\mu_t(\cdot|z)}{d\mu_t}(h) d\pi(z) \tag{23}$$

Eq. 23 is an extension of the Theorem 1 as detailed in Kerrigan et al. [29], and we provide the derivation in Appendix G. We note that $\mu_t(\cdot|z)$ is a Gaussian measure and can be expressed as $\mu_t(\cdot|z) = \mathcal{N}(m_t, C_t)$, with mean $m_t$ and trace-class covariance operator $C_t$. Inspired by Tong et al. [2], we choose $m_t$ and $C_t$ to have the following forms:

$$m_t = t \cdot h_1 + (1-t) \cdot h_0 \tag{24}$$

$$C_t = \sigma_{\min}^2 C_0 \tag{25}$$

where $C_0$ is the same Gaussian covariance operator defined for $\nu_0$ and $\sigma_{\min}$ is a small constant. Further, similar to finite-dimensional flow matching, we only consider the simplest vector field that applies a canonical transformation for Gaussian measures, such that the flow has the form: $\Phi_t(h_0|z) = m_t + \sigma_{\min} h_0 \approx t \cdot h_1 + (1-t) \cdot h_0$. From Eq. 1, we can get $\mathcal{G}_t(h|z) = h_1 - h_0$, indicating $\mathcal{G}_t(h|z)$ is independent of the time $t$ and the path from $h_0$ to $h_1$ is a direct, straight line. Equipped with well-constructed conditional vector field and probability measures, we can train a neural operator $\mathcal{G}_\theta$ with the conditional flow matching loss

$$\mathcal{L}_{\text{CFM}}^\dagger = \mathbb{E}_{t \sim \mathcal{U}[0,1], h \sim \mu_t, z \sim \pi(\nu_0, \nu_1)} \|\mathcal{G}_\theta(t, h) - \mathcal{G}_t(h|z)\|^2 \tag{26}$$

Next, we explore how to approximate the true optimal transport plan from optimal coupling of the joint measure $\pi(\nu_0, \nu_1)$. A common way for measuring the distance between two probability measure is 2-Wasserstein distance, which a special case of static Kantorovich formulation [50]. The static 2-Wasserstein distance is defined as follows

$$W_{\text{sta}}(\nu_0, \nu_1)_2^2 = \inf_{\pi \in \Pi} \int_{\mathcal{H} \times \mathcal{H}} \|h_0 - h_1\|^2 d\pi(h_0, h_1) \tag{27}$$

In the ODE framework, we also care about the dynamic form of the 2-Wasserstein distance to estimate the cost along the transport trajectory, which also is a special case of dynamic Kantorovich formulation [37].

$$W_{\text{dyn}}(\nu_0, \nu_1)_2^2 = \inf_{\mu_t, \mathcal{G}_t} \int_{\mathcal{H}} \int_0^1 \|\mathcal{G}_t(h)\|^2 d\mu_t(h) dt \tag{28}$$

Within the OFM framework, the marginal probability measure is a sum of Dirac measures as described in Eq. 22, and we selected $\nu_0$ as a Gaussian measure and assumed $\nu_1$ has full support on the Cameron-Martin space associated with $\nu_0$. Furthermore, the cost function of 2-Wasserstein distance is squared $L^2$ norm, which is continuous by nature. According to Theorem 4.3 and Lemma 4.4 of Chizat et al. [37], $W_{\text{sta}} = W_{\text{dyn}}$ for our specifically constructed $\mu_t$ and $\mathcal{G}_t$ in the sense of weak convergence. Therefore, to get the dynamic optimal transport plan, we only need to find a joint measure $\pi(\nu_0, \nu_1)$ that achieves the infimum in Eq. 27. In practice, we use a minibatch approximation of optimal coupling between $\nu_0$ and $\nu_1$. The above approach extends the dynamic (marginal) optimal transport framework of [2] to infinite-dimensional function space. The related work of Kerrigan et al. [45] addresses a similar problem, but from a different perspective. For a detailed comparison, please refer to Appendix Q.

# G Derivation of Eq. 23

In this part, we show the derivation of Eq. 23, which extends Theorem 1 of Kerrigan et al. [29]. The problem setting is given continuity equation and its weak form:

$$\int_0^1 \int_{\mathcal{H}} \frac{\partial \varphi(h,t)}{\partial t} + \langle \mathcal{G}_t(h), \nabla_h \varphi(h,t) \rangle d\mu_t(h) dt = 0, \quad \forall \varphi \in \mathrm{Cyl}(\mathcal{H} \times [0,1]) \tag{29}$$

we want to derive the following form of the conditional vector field under absolute continuity assumption and other mild conditions, where $z := (h_0, h_1) \in \mathcal{H} \times \mathcal{H}$.

$$\mathcal{G}_t(h) = \int_{\mathcal{H} \times \mathcal{H}} \mathcal{G}_t(h|z) \frac{d\mu_t(\cdot|z)}{d\mu_t}(h) d\pi(z) \tag{30}$$

First, $\int_0^1 \int_{\mathcal{H}} \frac{\partial \varphi(h,t)}{\partial t} d\mu_t(h) dt = \int_0^1 \int_{\mathcal{H}} \int_z \frac{\partial \varphi(h,t)}{\partial t} d\mu_t(h|z) d\pi(z) dt$. With continuity equation in strong form and the fact that $\mathcal{G}_t(h|z)$ induces $\mu_t(h|z)$ we have:

$$\int_0^1 \int_{\mathcal{H}} \int_z \frac{\partial \varphi(h,t)}{\partial t} d\mu_t(h|z) d\pi(z) dt = \int_0^1 \int_{\mathcal{H}} \int_z -\nabla \cdot (\varphi(h,t) \mathcal{G}_t(h|z)) d\mu_t(h|z) d\pi(z) dt$$

By the divergence-form identity:

$$\nabla \cdot (\varphi(h,t) \mathcal{G}_t(h|z)) = \langle \mathcal{G}_t(h), \nabla_h \varphi(h,t) \rangle + \varphi(h,t) \nabla \cdot \mathcal{G}_t(h|z)$$

Since we choose the smooth test function $\varphi(h,t)$ from $\mathrm{Cyl}(\mathcal{H} \times [0,1])$ and use the continuity equation in weak form, we assume term $\varphi(h,t) \nabla \cdot \mathcal{G}_t(h|z)$ disappears under integration. Thus we have

$$\int_0^1 \int_{\mathcal{H}} \frac{\partial \varphi(h,t)}{\partial t} d\mu_t(h) dt = -\int_0^1 \int_{\mathcal{H}} \int_z \langle \mathcal{G}_t(h|z), \nabla_h \varphi(h,t) \rangle d\mu_t(h|z) d\pi(z) dt$$

$$= -\int_0^1 \int_{\mathcal{H}} \int_z \langle \mathcal{G}_t(h|z), \nabla_h \varphi(h,t) \rangle \frac{d\mu_t(h|z)}{d\mu_t(h)} d\mu_t(h) d\pi(z) dt$$

$$= -\int_0^1 \int_{\mathcal{H}} \int_z \langle \mathcal{G}_t(h|z) \frac{d\mu_t(h|z)}{d\mu_t(h)}, \nabla_h \varphi(h,t) \rangle d\mu_t(h) d\pi(z) dt$$

$$= -\int_0^1 \int_{\mathcal{H}} \int_z \langle \mathcal{G}_t(h|z) \frac{d\mu_t(\cdot|z)}{d\mu_t}(h) d\pi(z), \nabla_h \varphi(h,t) \rangle d\mu_t(h) dt$$

$$= -\int_0^1 \int_{\mathcal{H}} \langle \int_z \mathcal{G}_t(h|z) \frac{d\mu_t(\cdot|z)}{d\mu_t}(h) d\pi(z), \nabla_h \varphi(h,t) \rangle d\mu_t(h) dt$$

On the other side, from Eq 29, we have

$$\int_0^1 \int_{\mathcal{H}} \frac{\partial \varphi(h,t)}{\partial t} d\mu_t(h) dt = -\int_0^1 \int_{\mathcal{H}} \langle \mathcal{G}_t(h), \nabla_h \varphi(h,t) \rangle d\mu_t(h) dt = 0, \quad \forall \varphi \in \mathrm{Cyl}(\mathcal{H} \times [0,1])$$

Thus $\mathcal{G}_t(h) = \int_z \mathcal{G}_t(h|z) \frac{d\mu_t(\cdot|z)}{d\mu_t}(h) d\pi(z) = \int_{\mathcal{H} \times \mathcal{H}} \mathcal{G}_t(h|z) \frac{d\mu_t(\cdot|z)}{d\mu_t}(h) d\pi(z)$

# H Derivation of Eq. 14

In this part, we show the detailed derivation of Eq. 14. In Flow Matching, the variable $z$ is chosen as a single data point from the coupling $\pi(u_0, u_1)$ where $u_1 \sim q_1$, and $u_0 \sim q_0 = \mathcal{N}(\mathbf{0}, K(\{x_1, x_2, \ldots, x_n\}))$. Considering the class of Gaussian conditional probability paths

$$p_t(u_t|z) = \mathcal{N}(u_t|m_t(z), \sigma_t(z)^2 K(\{x_1, x_2, \ldots, x_n\})) \tag{31}$$

With conditional flow $\phi_t(u_t|z) = \sigma_t u_0 + m_t$. Specially, we choose $m_t = t u_1 + (1-t) u_0$ and $\sigma_t = \sigma$, where $\sigma > 0$ is a small constant. From Eq. 1 (or Theorem 3 of Lipman et al. [11]), a vector that defines the Gaussian conditional flow is :

$$\mathcal{G}_t(u_t|z) = \frac{\sigma_t'}{\sigma_t}(u_t - m_t) + m_t'(u_1) \tag{32}$$

Then we can derive a closed-form expression for both the conditional probability and corresponding vector field [2] by plug in $\mu_t$ and $\sigma_t$ into Eq. 31 and Eq. 32

$$p_t(u_t|z) = \mathcal{N}(u_t|tu_1 + (1-t)u_0, \sigma^2 K(\{x_1, x_2, \ldots, x_n\})) \tag{33}$$

$$\mathcal{G}_t(u_t|u_1) = 0 + (u_1 - u_0) = u_1 - u_0 \tag{34}$$

Now, let's check the boundary conditions. At $t = 0$,

$$p_0(u_t|z) = \mathcal{N}(u_t|u_0, \sigma^2 K(\{x_1, x_2, \ldots, x_n\})) \xrightarrow{\sigma \to 0} \delta_{u_0} \tag{35}$$

At $t = 1$,

$$p_1(u_t|z) = \mathcal{N}(u_t|u_1, \sigma^2 K(\{x_1, x_2, \ldots, x_n\})) \xrightarrow{\sigma \to 0} \delta_{u_1} \tag{36}$$

From Eq. 9, we have $p_0(u_0) = \int p_0(u_t|z)\pi(z)dz = \int \delta_{u_0}\pi(u_0, u_1)du_0du_1 = q_0$ and $p_1(u_1) = \int p_1(u_t|z)\pi(z)dz = \int \delta_{u_1}\pi(u_0, u_1)du_0du_1 = q_1$, which show boundary conditions are satisfied.

# I   Proof of Proposition 3.1

**Proposition 3.1.** Given noisy observations $\{\widehat{u}(x_i)\}_{i=1}^n$, the posterior distribution is

$$\log \mathbb{P}\left(\{u(x_i)\}_{i=1}^m \middle| \{\widehat{u}(x_i)\}_{i=1}^n\right) = -\frac{\sum_{i=1}^n \|\widehat{u}(x_i) - u(x_i)\|^2}{2\sigma^2} + \log \mathbb{P}\left(\{u(x_i)\}_{i=1}^m\right) + C \tag{37}$$

Where the constant $C = -\frac{n}{2}\log(2\pi\sigma^2) - \log \mathbb{P}\left(\{\widehat{u}(x_i)\}_{i=1}^n\right)$.

*Proof.* With Bayes rule, we have:

$$\mathbb{P}\left(\{u(x_i)\}_{i=1}^m \middle| \{\widehat{u}(x_i)\}_{i=1}^n\right) = \frac{\mathbb{P}\left(\{\widehat{u}(x_i)\}_{i=1}^n \middle| \{u(x_i)\}_{i=1}^m\right) \cdot \mathbb{P}\left(\{u(x_i)\}_{i=1}^m\right)}{\mathbb{P}\left(\{\widehat{u}(x_i)\}_{i=1}^n\right)} \tag{38}$$

Taking the logarithm of Eq. 38, we have:

$$\log \mathbb{P}\left(\{u(x_i)\}_{i=1}^m \middle| \{\widehat{u}(x_i)\}_{i=1}^n\right) = \log \mathbb{P}\left(\{\widehat{u}(x_i)\}_{i=1}^n \middle| \{u(x_i)\}_{i=1}^m\right) + \log \mathbb{P}\left(\{u(x_i)\}_{i=1}^m\right) - \\ \log \mathbb{P}\left(\{\widehat{u}(x_i)\}_{i=1}^n\right) \tag{39}$$

Given $\epsilon_i \sim \mathcal{N}(0, \sigma^2)$ and $\{\epsilon_i\}_{i=1}^n$ is a multivariate Gaussian, then $\{\widehat{u}(x_i)\}_{i=1}^n \middle| \{u(x_i)\}_{i=1}^n$ is a shifted multivariate Gaussian with mean $\{u(x_i)\}_{i=1}^n$ translated from the original multivariate Gaussian $\{\epsilon_i\}_{i=1}^n$. Due to the translation invariance property of Gaussian distribution, We have :

$$\log \mathbb{P}\left(\{\widehat{u}(x_i)\}_{i=1}^n \middle| \{u(x_i)\}_{i=1}^n\right) = \log \mathbb{P}\left(\{\epsilon_i\}_{i=1}^n\right) = -\frac{\sum_{i=1}^n \|\widehat{u}(x_i) - u(x_i)\|^2}{2\sigma^2} - \frac{n}{2}\log(2\pi\sigma^2) \tag{40}$$

We notice $m > n$ and $\{\widehat{u}(x_i)\}_{i=1}^n$ only depends on $\{u(x_i)\}_{i=1}^n$, and doesn't depend on $\{u(x_i)\}_{i=n+1}^m$. Thus $\log \mathbb{P}\left(\{\widehat{u}(x_i)\}_{i=1}^n \middle| \{u(x_i)\}_{i=1}^m\right) = \log \mathbb{P}\left(\{\widehat{u}(x_i)\}_{i=1}^n \middle| \{u(x_i)\}_{i=1}^n\right)$.

For evaluating $\log \mathbb{P}\left(\{u(x_i)\}_{i=1}^m\right)$, which is the second part on the right-hand side of Eq. 39, we can efficiently calculate it with the trace estimator. The third part on the right hand side of Eq. 39 ($\log \mathbb{P}\left(\{\widehat{u}(x_i)\}_{i=1}^n\right)$) represents the evidence and is constant. Thus the posterior distribution of Eq 39 can be simplified as:

$$\log \mathbb{P}\left(\{u(x_i)\}_{i=1}^m \middle| \{\widehat{u}(x_i)\}_{i=1}^n\right) = -\frac{\sum_{i=1}^n \|\widehat{u}(x_i) - u(x_i)\|^2}{2\sigma^2} + \log \mathbb{P}\left(\{u(x_i)\}_{i=1}^m\right) + C \tag{41}$$

Where the constant $C = -\frac{n}{2}\log(2\pi\sigma^2) - \log \mathbb{P}\left(\{\widehat{u}(x_i)\}_{i=1}^n\right)$. $\qquad \square$

# J  Example of Posterior Samples

In this section, we initially present the regression result of OFM in another additional N-S scenario, as illustrated in Fig 7. Subsequently, we display more posterior samples used in the 2D regression examples. As depicted in Fig 9, 10, 11, OFM successfully generates realistic posterior samples that are consistent with the ground truth and demonstrate appropriate variability. In contrast, GP regression fails to produce explainable posterior samples.

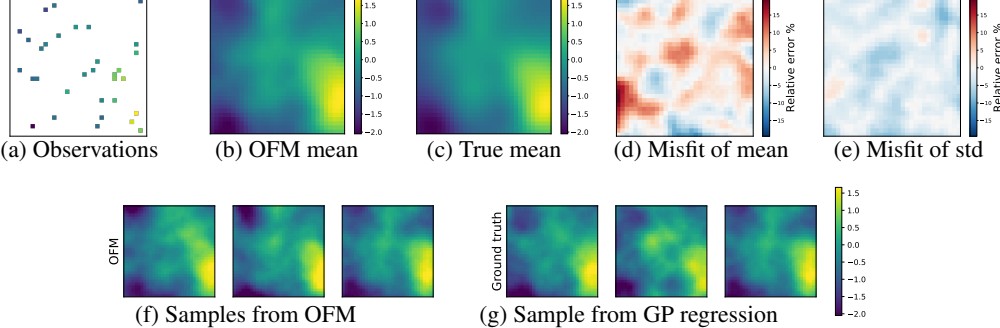

Figure 6: OFM regression on 2D GP data with resolution 32×32. (a) 32 random observations. (b) Predicted mean from OFM. (c) Ground truth mean from GP regression. (d) Misfit of the predicted mean. (e) Misfit of predicted standard deviation. (f) Predicted samples from OFM. (g) Predicted samples from GP regression.

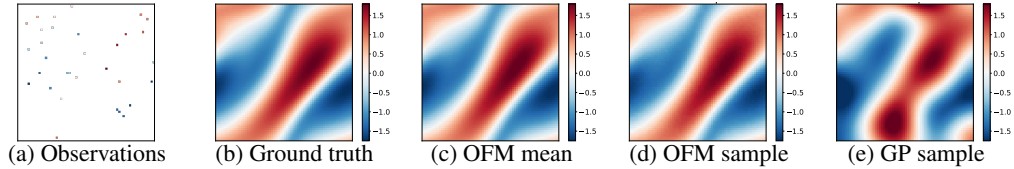

Figure 7: OFM regression on Navier-Stokes functional data with resolution $64 \times 64$. (a) 32 random observations. (b) Ground truth sample (c) Predicted mean from OFM. (d) One posterior sample from OFM. (e) One posterior sample from best fitted GP.

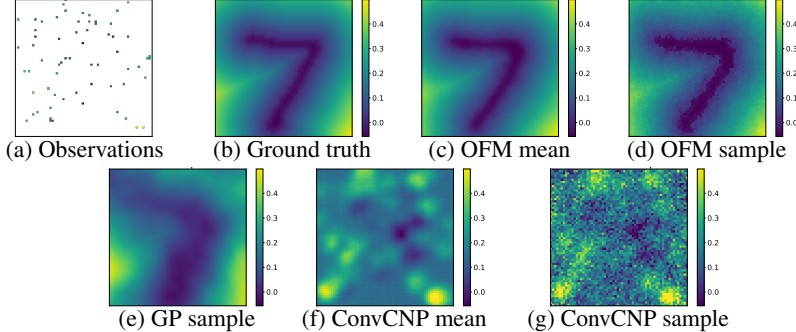

Figure 8: OFM regression on MNIST-SDF with resolution $64 \times 64$. (a) 64 random observations. (b) Ground truth sample. (c) Predicted mean from OFM. (d) One posterior sample from OFM. (e) One posterior sample from best fitted GP. (f) Predicted mean from ConvCNP. (g) One posterior sample from ConvCNP.

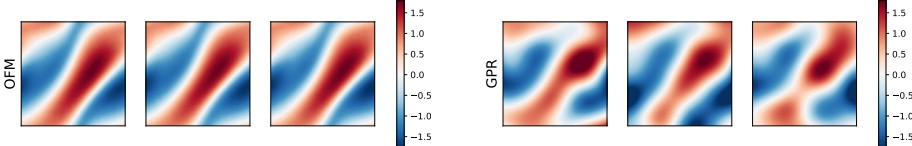

Figure 9: OFM regression on NS data. (**left**) Posterior samples from OFM. (**right**) Posterior samples from GP regression.

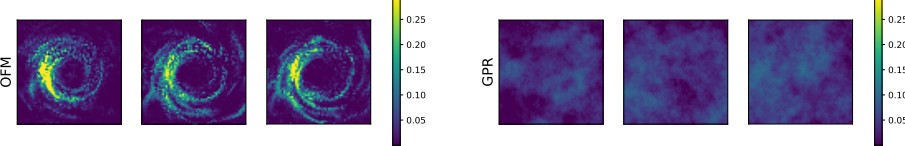

Figure 10: OFM regression on black hole data. (**left**) Posterior samples from OFM. (**right**) Posterior samples from GP regression.

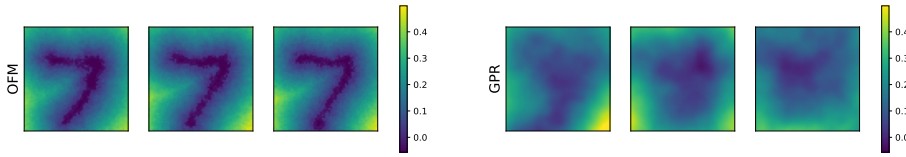

Figure 11: OFM regression on MNIST-SDF data. (**left**) Posterior samples from OFM. (**right**) Posterior samples from GP regression.

## K  Co-domain functional regression with OFM

In this section, we expand our regression framework to accommodate co-domain settings, as many function datasets feature a co-domain dimension greater than one. For example, earthquake waveform data commonly include three directional components, leading to a three-dimensional co-domain. Similarly, the velocity field in fluid dynamics usually features three directional components, also resulting in a dimension of co-domain of three.

We illustrate this extension through a 2D GP example with a co-domain of 3 (channel dimension of 3). In learning the prior, we define the reference measure ($\nu_0$) as a joint measure (Wiener measure) of three identical but independent Gaussian measures while the target measure ($\nu_1$) is another Wiener measure. We keep all other parameters unchanged as those described in the 2D GP regression tasks, with the only modification being an increase in the channel dimension from one to three. After training the prior (training detail provided in Appendix M), and provided 32 random observations across the three channels at co-locations, we then perform regression with OFM across these channels jointly. As demonstrated in Fig 12, OFM accurately estimate the mean and uncertainty across three channels.

## L  Posterior sampling with Stochastic Gradient Langevin Dynamics

In this section, we describe how to sample from posterior distribution with SGLD. We denote logarithmic posterior distribution (Eq. 41) as $\log \mathbb{P}_\theta$ and denote a set of posterior samples as $\{u_\theta^t\}_{t=1}^N$, where each $u_\theta^t$ is defined on a collection of point $\{x_i\}_{i=1}^m$.

By following the standard SGLD pipeline as described by Welling and Teh [4], we can obtain a set of $N$ posterior samples $\{u_\theta^t\}_{t=1}^N$. However, SGLD is known to be sensitive to the choice of regression parameters and can become trapped in local minima, leading to convergence issues, especially in regions of high curvature [51]. To mitigate these challenges, Shi et al. [1] proposed that within an invertible framework, drawing a posterior sample $u_\theta^t$ is equivalent to drawing a sample $a_\theta^t$ in Gaussian space, since $u_\theta^t$ uniquely defines $a_\theta^t$ and vice versa. This approach can stabilize the posterior sampling process and is less sensitive to the regression parameters due to the inherent smoothness of the Gaussian process. Additionally, Shi et al. [1] suggests starting from maximum a *posteriori*

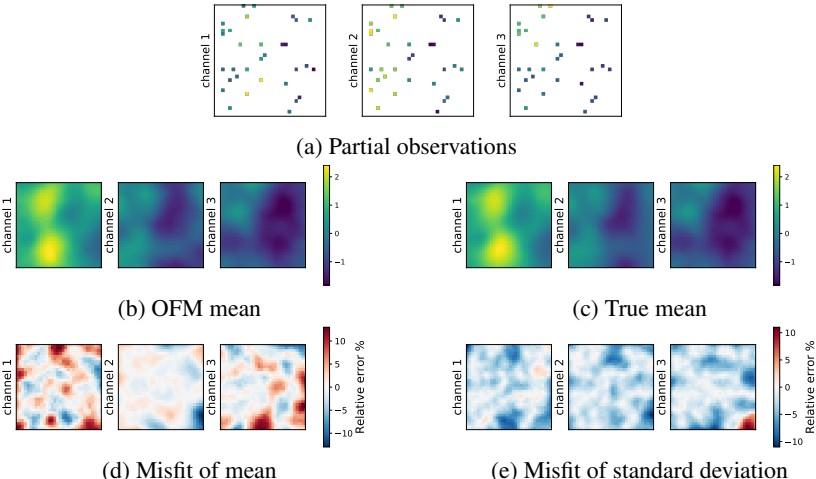

(a) Partial observations

(b) OFM mean

(c) True mean

(d) Misfit of mean

(e) Misfit of standard deviation

Figure 12: OFM regression on co-domain GP data with resolution 32x32. (a) 32 random observations at co-locations. (b) Predicted mean from OFM. (c) Ground truth mean from GP regression. (d) Misfit of the predicted mean. (e) Misfit of predicted standard deviation.

(MAP) estimate of $a_\theta^t$, denoted as $\overline{a_\theta}$, which can reduces the number of burn-in terations needed in SGLD. We adopt the same sampling strategy and the algorithm is reported in Algorithm 1

When the size of observations or context points $(\{\widehat{u}(x_i)\}_{i=1}^n)$ is 0, sampling from the posterior degrades to sampling from the prior, the results of which are presented in the subsequent section.

---

**Algorithm 1** Posterior sampling with SGLD

---

**Input and Parameters:** Logarithmic posterior distribution $\log \mathbb{P}_\theta$, temperature $T$, learning rate $\eta_t$, MAP $\overline{a}_\theta$, burn-in iteration $b$, sampling iteration $t_N$, total iteration $N$.

1: **Initialization**: $a_\theta^0 = \overline{a}_\theta$
2: **for** $t = 0, 1, 2, \ldots, N$ **do**
3:      Compute gradient of the posterior: $\nabla_{a_\theta} \log \mathbb{P}_\theta$
4:      Update $a_\theta^{t+1}$: $a_\theta^{t+1} = a_\theta^t + \frac{\eta_t}{2} \nabla \log \mathbb{P}_\theta + \sqrt{\eta_t T} \mathcal{N}(0, I)$
5:      **if** $t \geq b$ **then**
6:          Every $t_N$ iterations: obtain new sample $a_\theta^{t+1}$, and corresponding $u_\theta^{t+1}$
7:      **end if**
8: **end for**

---

## M   Prior learning with OFM

We now elaborate on the prior learning process and the corresponding performance evaluation. As shown in Algorithm 2, the training dataset is sampled from the unknown data measure $\nu_1$. Concretely, the training dataset consists of $M$ discretized functions $\{u_i|D_i\}_{i=1}^M$, where $u_i|D_i$ denotes a discretized observation of the $u_i$ function.

In practice, to simplify dataset preparation, one often uses the same discretization grid $D_i$ for all function samples, e.g. $D_i = \{x_1, \cdots, x_n\}$ regardless of the sample index "i". For the consistency of notions, let $h_0$ represents a batch of i.i.d discretized functions sampled from the training dataset (equivalently, sampled from $\nu_1$). Next, the reference Gaussian process $\nu_0 = \mathcal{N}(m_0, C_0)$ is known and determined by the user. With a slight abuse of notation, We choose to use notation $h_0, h_1$ for consistency purpose, in other parts of this paper, discretized $h_0, h_1$ is replaced with $a, u$ respectively.

For specific experiments setting, we employ Matern kernel to construct the reference GP and to prepare training datasets for 1D GP, 2D GP, and 1D TGP. We have set the kernel length $l = 0.01$ with a smoothness factor $\zeta = 0.5$ for all reference GPs. OFM maps the GP samples from reference GPs to data samples and is resolution-invariant, which means OFM can be trained with functions at any resolution and evaluated at any resolution.

**Algorithm 2** Learning a prior

---

**Input:** Reference Gaussian process $\nu_0 = \mathcal{N}(m_0, C_0)$, data measure $\nu_1$, batch size $b$, small constant $\sigma_{\min}$, discretized domain $D = \{x_1, \cdots x_n\}$

1: **while** Training **do**
2:     $h_0 \sim \nu_0; \quad h_1 \sim \nu_1$      # sample functions of size $b$ i.i.d from the measures on $D$
3:     $\pi \leftarrow \mathrm{OT}(h_0, h_1)$        # mini-batch optimal transport plan
4:     $(h_0, h_1) \sim \pi$
5:     $t \sim \mathcal{U}(0, 1)$
6:     $\mu_t \leftarrow t\, h_1 + (1 - t)\, h_0$
7:     $h_t \sim \mathcal{N}(\mu_t, \sigma_{\min}^2 C_0)$
8:     $\mathcal{L}_{\mathrm{CFM}}^{\dagger}(\theta) \leftarrow \left\| \mathcal{G}_\theta(t, x) - (h_1 - h_0) \right\|^2$
9:     $\theta \leftarrow \mathrm{Update}\big(\theta, \nabla_\theta \, \mathcal{L}_{\mathrm{CFM}}^{\dagger}(\theta)\big)$
10: **end while**
11: **return** $\mathcal{G}_\theta$

---

**1D GP dataset.** We choose $l = 0.3$ and $\zeta = 1.5$ and generate $20,000$ training samples on domain $[0, 1]$ with a fixed resolution of 256. We use autocovariance and histogram of point-wise value as metrics for evaluation. We evaluate OFM at several different resolutions shown Fig 13, 14, 15, which demonstrate OFM's excellent capability to learn the function prior.

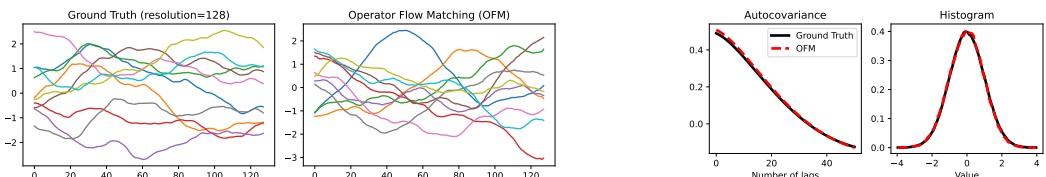

Figure 13: OFM for 1D GP prior learning, evaluated at resolution=128. (**left two**) Random samples from ground truth and generated by OFM. (**right two**) Autocovariance and histogram comparison

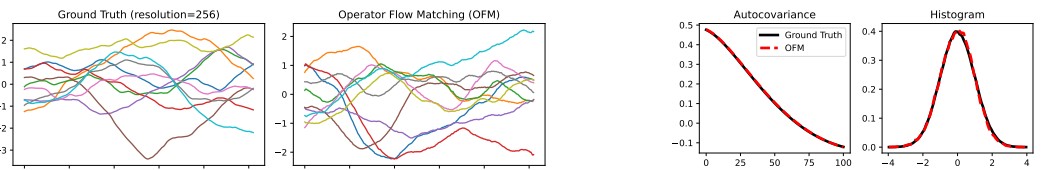

Figure 14: OFM for 1D GP prior learning, evaluated at resolution=256. (**left two**) Random samples from ground truth and generated by OFM. (**right two**) Autocovariance and histogram comparison

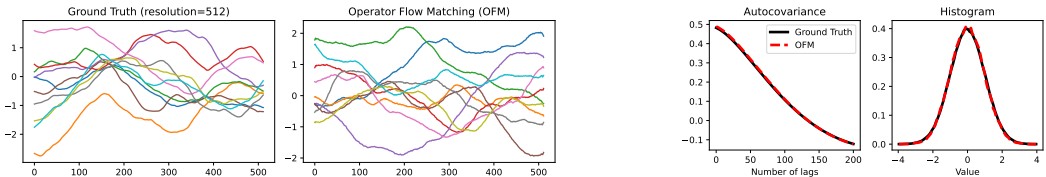

Figure 15: OFM for 1D GP prior learning, evaluated at resolution=512. (**left two**) Random samples from ground truth and generated by OFM. (**right two**) Autocovariance and histogram comparison

**1D TGP dataset.** We choose $l = 0.3$ and $\zeta = 1.5$ and generating $20,000$ training samples on domain $[0, 1]$ with a fixed resolution of 256. We set $[-1.2, 1.2]$ for the bounds. Results provided in Fig 16, 17, 18.

**2D Naiver-Stokes, Black hole, MNIST-SDF datasets.** All the following 2D datasets are defined on domain $[0, 1] \times [0, 1]$ and have a resolution of $64 \times 64$. We collected a 2D Navier-Stokes dataset

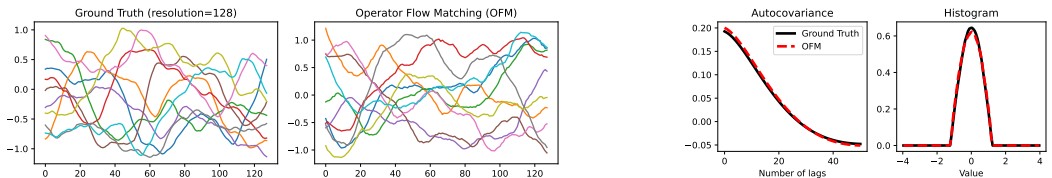

Figure 16: OFM for 1D TGP prior learning, evaluated at resolution=128. (**left two**) Random samples from ground truth and generated by OFM. (**right two**) Autocovariance and histogram comparison

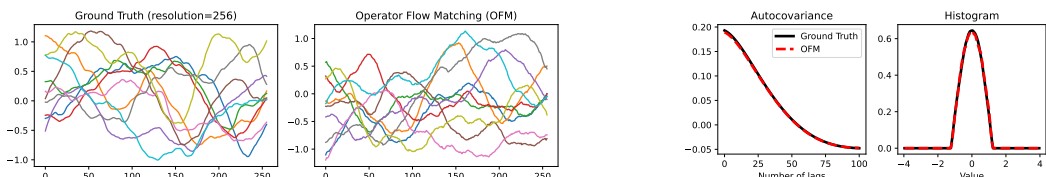

Figure 17: OFM for 1D TGP prior learning, evaluated at resolution=256. (**left two**) Random samples from ground truth and generated by OFM. (**right two**) Autocovariance and histogram comparison

consisting of 20000 samples, with viscosity $= 1e - 4$ . The results, including zero-shot super-resolution, are provided in Fig 19, 20. The learning of Black hole dataset, generated using expensive Monte Carlo method, is detailed in Fig 21, 22. Additionally, we trained OFM on $20,000$ MNIST-SDF samples, the outcomes are illustrated in Fig 23, 24.

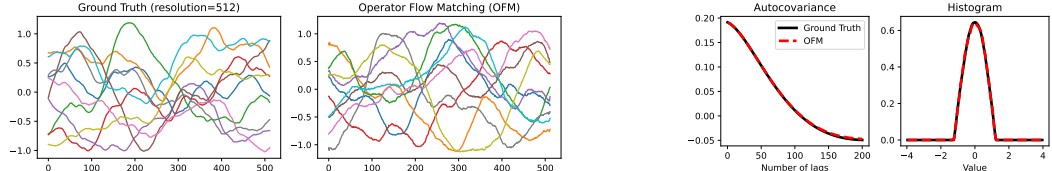

Figure 18: OFM for 1D TGP prior learning, evaluated at resolution=512. (**left two**) Random samples from ground truth and generated by OFM. (**right two**) Autocovariance and histogram comparison

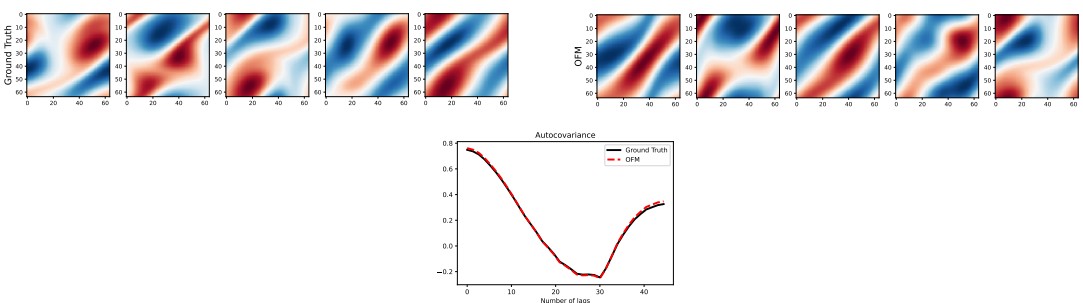

Figure 19: OFM for 2D N-S prior learning, evaluated at resolution=$64 \times 64$. (**top left**) Random samples from ground truth. (**top right**) Random samples generated by OFM. (**bottom**) Autocovariance comparison

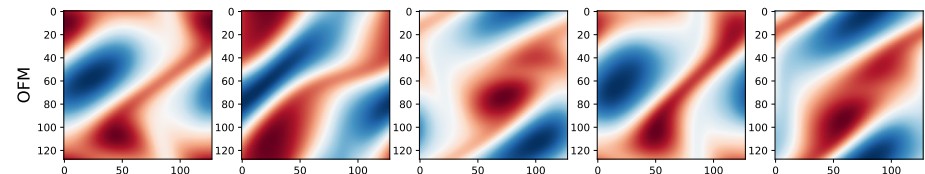

Figure 20: OFM for 2D N-S prior learning, evaluated at $128 \times 128$ resolution (zero-shot super-resolution)

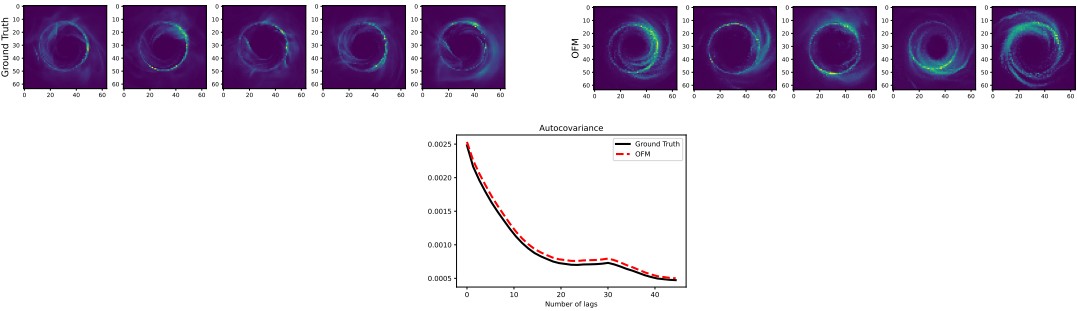

Figure 21: OFM for 2D black hole prior learning, evaluated at resolution=64. (**top left**) Random samples from ground truth. (**top right**) Random samples generated by OFM. (**bottom**) Autocovariance comparison

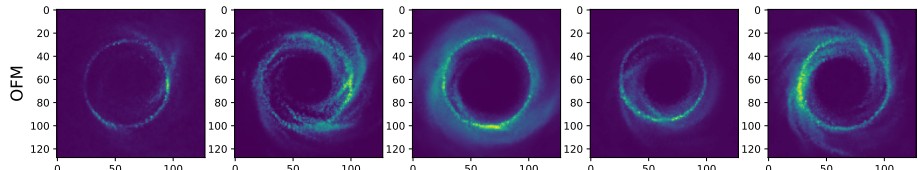

Figure 22: OFM for 2D black hole prior learning, evaluated at $128 \times 128$ resolution (zero-shot super-resolution)

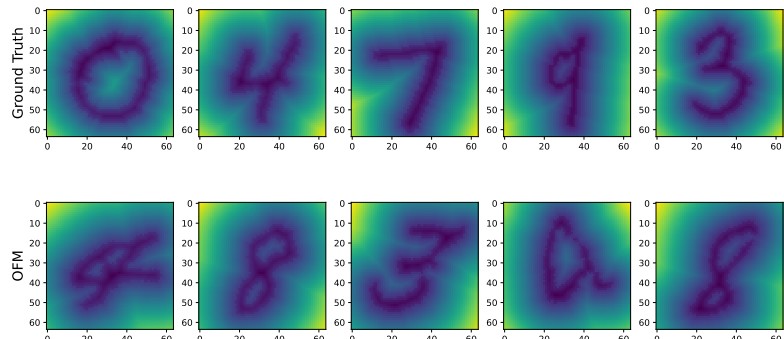

Figure 23: OFM for 2D MNIST-SDF prior learning, evaluated at $64 \times 64$ resolution. (**top**) Random samples from ground truth. (**bottom**) Random samples generated by OFM.

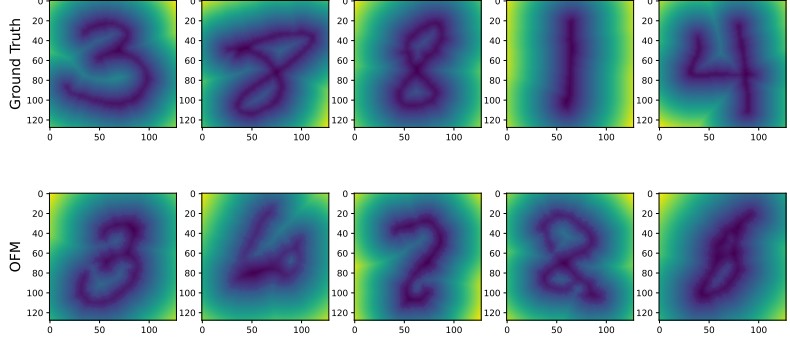

Figure 24: OFM for 2D MNIST-SDF prior learning, evaluated at $128 \times 128$ resolution. (**top**) Random samples from ground truth. (**bottom**) Random samples generated by OFM.

# N   Additional results of 1D GP regression with more complex kernels

In this section, we extend the 1D GP experiments by introducing more complex kernels for both data generation and regression, while holding all other settings identical to those used with the Matérn kernel. Specifically, we report results for the non-stationary Gibbs kernel and the Rational Quadratic (RQ) kernel.

For the Gibbs kernel, we use an input-dependent length-scale

$$\ell(x) = \ell_0 + \ell_1 x, \qquad x \in [0, 1],$$

which induces the covariance

$$k(x, x') = \sigma^2 \sqrt{\frac{2\, \ell(x)\, \ell(x')}{\ell(x)^2 + \ell(x')^2}} \, \exp\left( -\frac{(x - x')^2}{\ell(x)^2 + \ell(x')^2} \right).$$

In our setup we take $\ell_0 = 0.05$, $\ell_1 = 0.25$, and $\sigma = 1.0$.

For the RQ kernel, we use `sklearn.gaussian_process.kernels.RationalQuadratic` with length-scale $0.15$. As shown in Table 2, OFM consistently outperforms all baselines on these tasks.

Table 2: Comparison of OFM with baseline models on 1D GP with Gibbs kernel and RQK. Best performance in bold.

| Dataset → | 1D GP-Gibbs | | 1D GP-RQK | |
|---|---|---|---|---|
| Algorithm ↓ Metric → | SMSE | MSLL | SMSE | MSLL |
| NP | $4.0 \cdot 10^{-1}$ | $7.1 \cdot 10^{-1}$ | $5.8 \cdot 10^{-1}$ | $2.2 \cdot 10^{0}$ |
| ANP | $3.5 \cdot 10^{-1}$ | $6.2 \cdot 10^{-1}$ | $2.7 \cdot 10^{-1}$ | $2.1 \cdot 10^{0}$ |
| ConvCNP | $3.4 \cdot 10^{-1}$ | $1.1 \cdot 10^{-1}$ | $2.2 \cdot 10^{-1}$ | $6.9 \cdot 10^{-1}$ |
| DGP | $4.3 \cdot 10^{-1}$ | $8.6 \cdot 10^{-1}$ | $2.4 \cdot 10^{-1}$ | $1.2 \cdot 10^{0}$ |
| DSPP | $4.2 \cdot 10^{-1}$ | $7.1 \cdot 10^{-1}$ | $2.5 \cdot 10^{-1}$ | $4.2 \cdot 10^{-1}$ |
| OpFlow | $3.1 \cdot 10^{-1}$ | $6.9 \cdot 10^{-1}$ | $2.9 \cdot 10^{-1}$ | $5.0 \cdot 10^{-1}$ |
| **OFM(Ours)** | $\mathbf{2.9 \cdot 10^{-1}}$ | $\mathbf{8.7 \cdot 10^{-2}}$ | $\mathbf{2.1 \cdot 10^{-1}}$ | $\mathbf{9.4 \cdot 10^{-2}}$ |

# O   Details of experimental setup

In this section, we outline the details of experiments setup used in this paper. Since regression with OFM requires learning the prior first, we list the parameters used for learning the prior and regression separately. We employ FNO as the backbone, implemented using `neuraloperator` library [14]. All time reported in the subsequent tables are based on one computations performed using a single NVIDIA RTX A6000 (48 GB) graphics card. In all experiments, we use the `dopri5` ODE solver provided by `torchdiffeq` Chen et al. [52] with `atol=1e-5` and `rtol=1e-5`. Detailed posterior sampling algorithm is provided in Appendix L.

Table 3 details the parameters used for training the prior. For instance, in the 1D GP prior learning experiment, the dataset consists of 20,000 samples, each with a co-domain dimension (or channel) of one. The batch size is set at 1024, and the model is trained over 500 epochs. The total training time is about 0.76 hours, and the size of the trained model is 37.1 megabytes.

Tables 4, 5, and 6 detail the parameters for SGLD sampling as described in Algorithm 1. For example, in the 1D GP regression experiment, the regression takes 40,000 iterations with a burn-in phase of 3,000 iterations. Posterior samples are collected every 10 iterations. The temperature for the injected noise during the gradient update is set at 1, and the learning rate decays exponentially from 0.005 to 0.004 (defined in Algorithm 1). We average 32 runs with the Hutchinson trace estimator to evaluate the likelihood, utilizing GPU parallel computing. The noise level, as specified in Equation 41, is 0.01 in this regression task. Then given 6 random observations, we ask for the posterior samples across 128 points. The GPU memory usage for the regression task is 4 gigabytes, with the total runtime to 4.91 hours.

Table 3: Parameters used in experiments of prior learning

| Datasets | Size of Dataset | Channels | Batch Size | Epochs | Training Time | Model Size |
|---|---|---|---|---|---|---|
| 1D GP | $2 \cdot 10^4$ | 1 | 1024 | $5 \cdot 10^2$ | 0.76 h | 37.1 MB |
| 1D TGP | $2 \cdot 10^4$ | 1 | 1024 | $5 \cdot 10^2$ | 1.24 h | 37.1 MB |
| 2D GP | $2 \cdot 10^4$ | 1 | 256 | $5 \cdot 10^2$ | 1.14 h | 76 MB |
| 2D co-domain GP | $2 \cdot 10^4$ | 3 | 256 | $5 \cdot 10^2$ | 1.01 h | 76 MB |
| 2D N-S | $2 \cdot 10^4$ | 1 | 256 | $5 \cdot 10^2$ | 3.79 h | 286 MB |
| 2D Black hole | $1.2 \cdot 10^4$ | 1 | 256 | $5 \cdot 10^2$ | 2.28 h | 286 MB |
| 2D MNIST-SDF | $2 \cdot 10^4$ | 1 | 256 | $5 \cdot 10^2$ | 8.31 h | 286 MB |

Table 4: Parameters used in regression experiments - Part A

| Datasets | Total Iteration | Burn-in Iteration | Sampling Iterations | Temperature of Noise |
|---|---|---|---|---|
| 1D GP | $4 \cdot 10^4$ | $3 \cdot 10^3$ | 10 | 1 |
| 1D TGP | $4 \cdot 10^4$ | $3 \cdot 10^3$ | 10 | 1 |
| 2D GP | $2 \cdot 10^4$ | $3 \cdot 10^3$ | 10 | 1 |
| 2D co-domain GP | $2 \cdot 10^4$ | $3 \cdot 10^3$ | 10 | 1 |
| 2D N-S | $2 \cdot 10^4$ | $3 \cdot 10^3$ | 10 | 1 |
| 2D Black hole | $2 \cdot 10^4$ | $3 \cdot 10^3$ | 10 | 1 |
| 2D MNIST-SDF | $2 \cdot 10^4$ | $3 \cdot 10^3$ | 10 | 1 |

# P Ablation and scaling studies for mini-batch optimal transport and Hutchinson trace estimator

We first present an ablation study of the optimal transport plan. In this study, we revisit prior learning on the N-S dataset by training two models: one using independent coupling and the other employing a mini-batch optimal transport plan. Both models were trained with a batch size of 64. For evaluation, we compare the mean squared error (MSE) of density, autocovariance and spectral characteristics between 1,000 real and generated samples. Additionally, we report the convergent training loss (squared $L^2$ loss) and the number of function evaluations (NFE) required for sampling with adaptive ODE solver (dopri5). As shown in Table 7, the mini-batch OT plan outperforms the independent coupling approach in terms of pointwise accuracy, spectral fidelity, and convergent $L^2$ loss, while also requiring fewer NFE and enabling faster sampling. Our findings indicate that as the mini-batch size increases, the model learns the prior with reduced error. This improvement may be attributed to the mini-batch optimal transport plan approaching the true optimal transport plan with larger batch sizes.

Next, we explore the variance of the Hutchinson trace estimator in likelihood estimation with a scaling experiment. For this experiment, we use a 1D GP example with parameters described in Section M and a resolution of 128. We randomly draw a GP sample from the prior and evaluate the log likelihood by integrating the divergence as shown in Eq. 15, while also estimating it using the Hutchinson trace estimator from Eq. 16. In this scaling experiment, the number of noise sample ($n_{\text{noise}}$) for the Hutchinson estimator is set to $(4, 8, 16, 32, 64, 128)$. For each $n_{\text{noise}}$, we repeat the experiment 100 times and report the mean and standard deviation of the predicted likelihood. The exact likelihood is computed by directly integrating the trace of the Jacobian. We repeat this procedure for 5 different random 1D GP samples. As reported in Table 8, the standard deviation of the Hutchinson trace estimator decreases rapidly as $n_{\text{noise}}$ increases, and the predicted means always align closely with the ground truth.

Furthermore, even at smaller $n_{\text{noise}}$ values, where a relative larger variance is expected, the performance of the posterior sampling appears robust (see Table 5). We hypothesize that this is due to stochastic nature of posterior sampling algorithm (SGLD), which requires injected perturbations, renders its performance relatively insensitive to the choice of $n_{\text{noise}}$.

Table 5: Parameters used in regression experiments - Part B

| Datasets | Initial Learning Rate | End Learning Rate | Hutchinson Samples | Noise Level |
|---|---|---|---|---|
| 1D GP | $5 \cdot 10^{-3}$ | $4 \cdot 10^{-3}$ | 32 | $1 \cdot 10^{-2}$ |
| 1D TGP | $5 \cdot 10^{-3}$ | $4 \cdot 10^{-3}$ | 32 | $1 \cdot 10^{-3}$ |
| 2D GP | $1 \cdot 10^{-3}$ | $8 \cdot 10^{-4}$ | 32 | $1 \cdot 10^{-2}$ |
| 2D co-domain GP | $1 \cdot 10^{-3}$ | $8 \cdot 10^{-4}$ | 16 | $1 \cdot 10^{-2}$ |
| 2D N-S | $3 \cdot 10^{-3}$ | $2 \cdot 10^{-3}$ | 8 | $1 \cdot 10^{-3}$ |
| 2D Black hole | $5 \cdot 10^{-3}$ | $4 \cdot 10^{-3}$ | 8 | $1 \cdot 10^{-3}$ |
| 2D MNIST-SDF | $5 \cdot 10^{-3}$ | $4 \cdot 10^{-3}$ | 8 | $1 \cdot 10^{-3}$ |

Table 6: Parameters used in regression experiment - Part C

| Datasets | Number of Observations | Inquired Grids | GPU Memory | Running Time |
|---|---|---|---|---|
| 1D GP | 6 | 128 | 4 GB | 4.91 h |
| 1D TGP | 3 | 128 | 4 GB | 5.42 h |
| 2D GP | 32 | $32 \times 32$ | 22 GB | 9.70 h |
| 2D co-domain GP | 32 | $32 \times 32$ | 31 GB | 5.05 h |
| 2D N-S | 32 | $64 \times 64$ | 44 GB | 13.65 h |
| 2D Black hole | 32 | $64 \times 64$ | 44 GB | 13.37 h |
| 2D MNIST-SDF | 64 | $64 \times 64$ | 44 GB | 9.41 h |

Table 7: Scaling study for the size of mini-batch given optimal transport plan, best performance in bold

| Metrics | Density-MSE | Autocovariance-MSE | Spectra-MSE | Convergent $L^2$ Loss | NFE |
|---|---|---|---|---|---|
| Independent | $3.4 \cdot 10^{-5}$ | $7.4 \cdot 10^{-5}$ | $1.3 \cdot 10^{1}$ | $5.1 \cdot 10^{-2}$ | 168 |
| mini-batch = 32 | $3.0 \cdot 10^{-5}$ | $1.3 \cdot 10^{-4}$ | $5.0 \cdot 10^{0}$ | $2.3 \cdot 10^{-2}$ | **141** |
| mini-batch = 64 | $\mathbf{9.8 \cdot 10^{-6}}$ | $9.8 \cdot 10^{-5}$ | $6.3 \cdot 10^{0}$ | $1.9 \cdot 10^{-2}$ | 153 |
| mini-batch = 128 | $2.4 \cdot 10^{-5}$ | $\mathbf{1.7 \cdot 10^{-5}}$ | $\mathbf{3.6 \cdot 10^{0}}$ | $\mathbf{1.5 \cdot 10^{-2}}$ | 182 |

Table 8: Scaling study for the number of noise samples of Hutchinson trace estimator

| | $n_{\text{noise}} = 4$ | $n_{\text{noise}} = 8$ | $n_{\text{noise}} = 16$ | $n_{\text{noise}} = 32$ | $n_{\text{noise}} = 64$ | $n_{\text{noise}} = 128$ | exact |
|---|---|---|---|---|---|---|---|
| sample 1 | $-481.7 \pm 9.4$ | $-482.9 \pm 7.7$ | $-480.6 \pm 4.9$ | $-481.4 \pm 3.8$ | $-481.1 \pm 2.5$ | $-480.8 \pm 1.8$ | $-482.7$ |
| sample 2 | $-482.8 \pm 9.8$ | $-481.8 \pm 6.7$ | $-481.3 \pm 4.9$ | $-480.8 \pm 3.5$ | $-481.1 \pm 2.7$ | $-481.2 \pm 1.7$ | $-482.0$ |
| sample 3 | $-479.7 \pm 10.2$ | $-479.7 \pm 7.6$ | $-478.2 \pm 4.7$ | $-478.6 \pm 3.4$ | $-478.8 \pm 2.6$ | $-478.8 \pm 1.7$ | $-479.1$ |
| sample 4 | $-476.9 \pm 10.0$ | $-477.0 \pm 6.6$ | $-477.9 \pm 5.6$ | $-478.5 \pm 4.0$ | $-478.1 \pm 2.7$ | $-478.0 \pm 1.8$ | $-477.4$ |
| sample 5 | $-479.4 \pm 10.7$ | $-479.2 \pm 6.6$ | $-479.2 \pm 5.3$ | $-479.4 \pm 3.6$ | $-479.6 \pm 2.8$ | $-479.3 \pm 2.0$ | $-478.5$ |

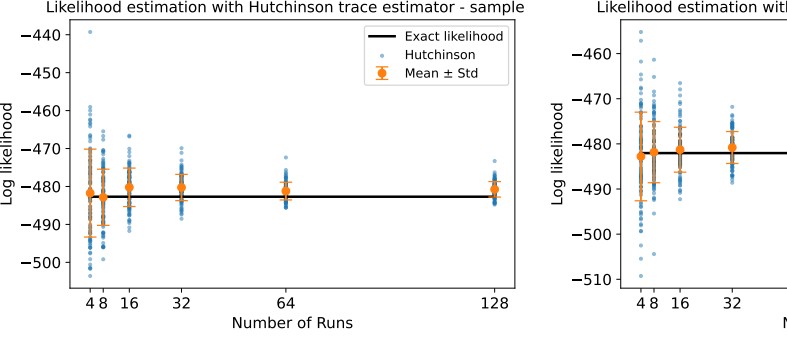

Figure 25: Log likelihood by the integration of divergence, (**left**) plot for first GP sample. (**right**) plot for second GP sample

# Q  Detailed analysis of OFM and comparison with existing methods

In this section, we elaborate the connection and difference with pervious work, highlight contributions and potential limitations of our work. The regression with OFM involves a two-steps process: (i) learning a prior on function space, and (ii) sampling from the posterior given observations. Consequently, the OFM framework has connections with both generative models on function space and the models developed for functional regression. In the following, we provide a comprehensive comparative analysis with related models and baselines, including operator flow (OPFLOW) [1], conditional optimal transport flow matching (COT-FM) [45], conditional models (NPs) [7, 34]

**Comparison with OPFLOW.** OPFLOW introduces invertible neural operators, which generalizes RealNVP [53] to function space and maps any collection of points sampled from a GP to a new collection of points in the data space, using the maximum likelihood principle [1]. This method captures the likelihood of any collection of point consistently as the resolution increases and allows for UFR using SGLD. Despite these advantages, the requirement for an invertible neural operator brings training and expressiveness challenges. To be specific, OPFLOW failed on all non-GP regression tasks in this paper because the prior learning stage suffered from mode collapse during training. On the contrary, OFM adopts a simulation-free ODE framework for prior learning, which offers enhanced expressiveness and ensures training stability through a simple regression objective while avoiding using the invertible neural operator. In addition, OFM proposes a non-trivial extension of UFR to the simulation-free ODE framework. These improvements render OFM a more practical solution for challenging functional regression tasks.

**Comparison with COT-FM.** COT-FM [45] proposes a conditional generalization of Benamou-Brenier Theorem [54], formulating a conditional optimal transport plan that applicable for both Euclidean and Hilbert space. In contrast, OFM employs an unconditional optimal transport plan in Hilbert space based on dynamic Kantorovich formulation, which is initially generalized for unbalanced optimal transport [37]. The advantage of COT-FM lies in its ability to flexibly incorporate specific conditions tailored for conditional generative tasks. However, COT-FM is not suitable for functional regression tasks due to: (i) COT-FM is contingent upon both the reference and target being influenced by conditions, and the vector field learnt is triangular, designed to transport jointly the coupling of a reference measure and a condition measure. In UFR setting, the learnt prior is required to be unconditioned, (ii) the coupling with condition measure typically prevents inducing valid stochastic process, even when the reference measure is a Gaussian measure, (iii) cannot provide point evaluation of probability density. Last, We should notice, the development of OFM is different and independent of COT-FM, the former with a focus on stochastic process learning and Bayesian functional regression.

**Comparison with conditional models.** NPs were developed to address the computational and restrictive prior challenges of Gaussian Processes, utilizing neural networks for efficiency [7]. However, several recent studies have discussed the drawbacks in the formulation of NPs, raising concerns that NPs might not learn the underlying function distribution [1, 22, 55].

Notably, NPs treats the point cloud data as a set of values, ignoring the metric space of the data [55]. This can lead to misinterpretations of a function sampled at different resolutions as distinct functions (Appendix A.1 of [22]). Furthermore, NPs rely on encoding input data into finite-dimensional, Gaussian-distributed latent variables before projecting these into an infinite-dimensional space. This process tends to lose consistency at higher resolutions. Moreover, the Bayesian regression framework underpinning NPs focuses on point sets rather than the functions themselves, leading to a dilution of prior information with increasing data points.

In recent study, diffusion-based variants of NPs (NDP) [34], was proposed to leverage the expressiveness of diffusion models [24, 25]. Nonetheless, the formulation of NDP does not address the aforementioned issues of NPs and introduces two more problems: (i) NDP fails to induce a valid stochastic process as it does not satisfy the marginal consistency criterion required by Kolmogorov Extension Theorem [39], and (ii) it relies on uncorrelated Gaussian noise for denoising, which is not applicable in function spaces [26]. Oppositely, OFM establishes a more theoretically sound framework by rigorously defining learning within function spaces. Additionally, Bayesian functional regression within the OFM framework adheres to valid stochastic processes, offering a robust and theoretically grounded solution. Last, we provide a high-level comparison of OFM and NP as shown in Table 9.

Table 9: High-level comparison of OFM and NP.

| Aspect | OFM | Neural Processes (NP) |
|---|---|---|
| Modeling target | Global *prior* over stochastic processes with a valid joint density. | Amortized *conditional* model without explicit tractable joint density over full functions. |
| Arbitrary queries | Set-size / location agnostic; evaluate on any grid or point set. | Set-size / location agnostic; trained with random context/target splits. Performance may degrade under extreme sparsity or distribution shift |
| Uncertainty & likelihoods | Tractable log-densities and posterior sampling enable principled Bayesian inference. | Predictive densities are available and trained via conditional likelihood; no closed-form function-level likelihood is typically specified. Uncertainty reflects model/context coverage and may be miscalibrated. |
| Structure & extrapolation | Captures global geometry and non-stationarity via the learned flow. | Learns inductive biases from data through context summaries; strong interpolation, but extrapolation and long-range generalization are not guaranteed and depend on training distribution. |

**Limitations.** Despite these advances, the current regression framework with OFM is primarily limited to low-dimensional data (1D and 2D in this study). This limitation stems from the challenges associated with learning operators for functions defined on high-dimensional domains—an area that remains underdeveloped both computationally and in terms of dataset availability [15]. Additionally, while the time complexity for regression with OFM is $\mathcal{O}(m^2)$, the incorporation of additional components significantly increases its computational resource requirements compared to classical GP regression.

There are several potential paths to mitigate the computation concerns: (i) Carefully adjusting the step size $(\eta_t)$ and temperature $(T)$ in the SGLD algorithm to prevent overly large gradient updates. (ii) Running the trace estimator on multiple GPUs, use mixed-precision FNO [43], or adopt efficient neural operators [56]. (iii) Using more efficient ODE solvers. Our current implementation uses a high-order ODE solver (dopri5) that requires over 100 steps for sampling. Switching to a more modern and efficient solver, such as the DPM-Solver [57], could potentially reduce the number of evaluations, freeing up significant GPU resources.

