# OpenReview forum: "Stochastic Process Learning via Operator Flow Matching"
_NeurIPS.cc/2025/Conference — NeurIPS 2025 spotlight_

### Official Review · Reviewer_sYHe · 2025-06-29

**Clarity:** 2
**Significance:** 4
**Originality:** 3
**Rating:** 5
**Confidence:** 4

**Summary:**

The authors propose a new framework, called Operator Flow Matching (OFM), for learning stochastic processes on function spaces. In brief, the authors extend flow-based generative modelling via Fourier neural operators (FNO) to map a reference Gaussian process (GP) to an unknown target process. Formally, they define a continuous-time ODE on a Hilbert space that “flows” samples from a simple GP prior ($\nu_0$) to the data distribution ($\nu_1$). By parameterizing the time-dependent vector field $G_t$ with a neural operator $G_\theta(t,\cdot)$, they can (in principle) learn a bijection between function spaces. This yields an explicit transport map so that any finite collection of points from the learned process has a tractable probability density under OFM. Once trained, the model can perform Bayesian regression: given partial noisy observations of a function, it can compute the posterior over the function values at new points (via Bayes’ rule and stochastic gradient Langevin dynamics (SGLD) in the latent GP space). Several claimed contributions are highlighted: (1) Novelty: extending flow-matching or stochastic interpolant methods to infinite-dimensional stochastic processes via operator learning; (2) Unifying prior and posterior inference: providing a unified framework for prior sampling and exact (tractable) likelihood estimation, enabling “universal functional regression” (UFR) with learned priors; (3) Exact densities: learning an exact (invertible) mapping so that the model supplies the true prior and posterior densities over any finite evaluation of the process. Empirically, OFM is tested on several synthetic datasets (1D/2D Gaussian processes, truncated GPs, and highly non-Gaussian processes from Navier–Stokes equations, simulated black-hole data, MNIST-SDF fields).

**Questions:**

NA

**Ethical Concerns:**

["NO or VERY MINOR ethics concerns only"]

**Final Justification:**

I keep my positive rating, voting for the clear acceptance.

**Limitations:**

Neither limitations of the proposed framework and its theoretical analysis, nor negative impact have been discussed in the main body. The high computational cost is hinted in the conclusion, but I feel that this deserves more serious report and discussion in the main body rather than in Appendix N.

**Paper Formatting Concerns:**

I didn't detect any formatting issues.

**Quality:**

3

**Strengths And Weaknesses:**

## Strengths

### __1) Integration of flows and neural operators:__

The paper introduces an original approach by combining flow-matching models with neural operators. By treating functions as first-class objects, OFM learns an infinite-dimensional transport map, which is innovative. This method naturally extends normalizing flows and stochastic interpolant models from finite-dimensional data to functional data. The authors provide a clear formulation of the continuous-time flow on a Hilbert space, showing how a diffeomorphism $\Phi_t$ pushes a reference GP measure to the data measure. The use of a Fourier neural operator (FNO) to parametrize the vector field $G_\theta(t,h)$ is appropriate to ensure resolution agnosticism (the model can handle discretized functions at arbitrary grids).

### __2) Mathematically grounded:__

The framework is built on solid theoretical ideas from optimal transport and infinite-dimensional analysis. The authors derive a continuity equation for the time-varying measure path and explain how the log-density can be obtained by integrating the divergence of $G_t$. They further leverage Kantorovich’s formulation of dynamic optimal transport together with a linear interpolation strategy for Gaussian measures to obtain a simple vector field that couples a reference GP sample $h_0$ with a target sample $h_1$. While I am not expert in these topics, I find that these constructions are internally consistent and grounded in prior work (e.g. Chizat et al. 2018, Tong et al. 2023). In short, the method is not ad-hoc: it extends finite-dimensional flow matching to function spaces in a principled way.

### __3) Tractable likelihoods and UFR:__

A key strength is that OFM can compute exact log-likelihoods for any finite set of function values. Once $G_\theta$ is trained, one can evaluate $\log p(u_1)$ for a new function evaluation $u_1$ by integrating as in Eq. (15). The paper carefully notes the $\mathcal{O}(m^2)$ cost of divergence computation and employs the Hutchinson estimator to make it $\mathcal{O}(m)$. This allows the use of Bayes’ rule for regression. In practice, the authors draw posterior samples via SGLD in the GP input space, then map them to function space. This gives a “universal functional regression” (UFR) procedure that generalizes GP regression to arbitrary learned priors. I find that the fact that OFM retains exact densities (unlike prior implicit models) is a strong advantage.

### __4) Empirical evidence:__

The experimental results are compelling. On synthetic GP tasks (1D and 2D), OFM matches or surpasses classical GP performance, and on more challenging non-Gaussian tasks it significantly outperforms baselines. The paper' supplementary material includes ablation and scaling studies to justify design choices, and also provides detailed architecture and training parameters. Moreover, the authors emphasize reproducibility: the code and logs are provided.



## Weaknesses

### 1) Dense presentation and clarity:

While I acknowledge that writing operator learning papers for general ML audience is particularly challenging due to high-level of abstraction and mathematical interaction of diverse fields, I feel that the paper is mathematically dense, which may hinder its accessibility. Many readers may find the notation and measure-theoretic language heavy. For example, Section 3 immediately introduces pushforward measures on a Hilbert space and continuity equations, which can be difficult to parse. While the technical content is correct, the exposition is terse: key concepts like Gaussian measures on function spaces or the Cameron–Martin condition are assumed known. The authors do provide Appendix B–D as “foundational overview”, but the main narrative would benefit from more intuition. The introduction of the conditional flow matching in Sec. 3.2 is similarly rapid, with many equations in succession. Since NeurIPS audience is broad; more high-level guidance or illustrative text would improve clarity (e.g. a step-by-step summary of the two-phase algorithm in plain language). I am aware that this is not an easy task due to the limited length of the main body, but I would still strongly encourage authors to improve on this aspect.

### __2) Novelty relative to recent work:__

The paper claims to be _the first to extend flow matching to stochastic processes__. However, it cites a recent paper (Ref [29]: “Functional Flow Matching” by Kerrigan et al., 2023) that appears to do something very similar. Indeed, the related work explicitly notes that [29] generalizes flow matching to function spaces. The authors argue that prior works (including [29]) “do not support UFR” or density evaluation, but this distinction should be made clearer. If [29] already develops the theory of infinite-dimensional flows, the novelty here is partly in combining it with operator networks and Bayesian regression. It would strengthen the paper to more explicitly state __“_unlike [29], we can compute densities and do conditional sampling_,”__ perhaps with a succinct comparison. As it stands, __the originality claim needs nuance__: OFM is novel in its application and integration (and outperforms [29] presumably), but flow-in-function-space per se is not entirely new.

### __3) Theoretical assumptions and scope:__

The method relies on strong assumptions that deserve discussion. For instance, the reference measure $\nu_0$ is a GP with some covariance operator, and $\nu_1$ is assumed to be absolutely continuous with respect to $\nu_0$ (so that an invertible transport exists). In practice this means the true process must lie in the Cameron–Martin space of the GP prior. The paper does not discuss what happens if this is violated. Also, computing a flow on infinite dimensions typically requires technical conditions (e.g. continuity of $G_t$, compactness of support, etc.). While the appendices address these issues, it would be helpful to acknowledge in the main text any limitations (e.g. “we assume the target measure has full support on the same space as the GP”) and what that implies. Relatedly, the choice of a linear interpolation for Gaussian measures is convenient, but may not be optimal for complex target distributions. To me it is unclear how the approach handles non-Gaussian $\nu_1$ during training, since the conditional flow matching still assumes $\mu_t(\cdot|z)$ is Gaussian. If $\nu_1$ is far from Gaussian, the learned map may suffer. The empirical results suggest it works, but a discussion of this modeling assumption would strengthen the theory section.

### __4) Empirical evaluation details:__

While the results are promising, some experimental questions remain:
- (a) Statistical robustness: The reported results (e.g. Table 2) do not include confidence intervals or multiple-trial averages. It is unclear how sensitive OFM is to random initialization or data sampling.

- (b) Baselines and fairness: The baselines chosen (GP, deep GP, NP family, “OPFlow”[1]) are reasonable, but other recent methods could be considered. For example, Neural Processes with latent flows or diffusion-based process models (e.g. [34] Neural Diffusion Processes) might be relevant, although those are mostly unconditional.

- (c) Computational cost: The paper notes in passing that “OFM demands more computational resources”, but does not quantify it in the main text. From Appendix tables we see training can take many GPU-hours and memory (dozens of GB). This could limit applicability, especially on high-resolution 2D/3D domains. Including some runtime/memory comparison or at least discussing scalability would be useful.

### __5) Lack of ablation in main text:__

The contributions bullet promises _“extensive ablation and scaling studies”_, but no ablation figures appear in the main body. It would help to include at least one ablation table or figure (for example, the effect of using Hutchinson estimation vs exact divergence, or the impact of neural operator depth) in the paper, rather than hiding all such results in appendices. This would boost confidence that each component (e.g. neural operator architecture, trace estimator, SGLD sampling) is necessary and correctly implemented.

### __6) Evaluation on realistic tasks:__

All experiments are on synthetic or simulation-based data. While this is acceptable for a methods paper, testing on at least one real-world dataset (e.g. geospatial data, time series, etc.) would demonstrate broader applicability. Also, the noise model is fixed to i.i.d. Gaussian in all cases; exploring non-i.i.d. or non-Gaussian noise could be informative, since many applications (e.g. cosmology, climate) have complex noise.

---

> ### Author Rebuttal · Authors · 2025-07-31
>
> We appreciate the reviewer for the thorough review, detailed summary, and constructive suggestions. We provide our responses below
>
> **W1. Dense presentation and clarity**
>
> We appreciate the reviewer’s observation that presenting a rigorous mathematical framework spanning several disciplines within the space constraints of the main text is challenging.
>
> To further improve clarity for readers from diverse backgrounds, we have (i) added an appendix section covering neural operators (operator learning), flow matching, Gaussian measures on function spaces, and the Cameron–Martin theorem to provide readers with essential background; and (ii) included additional explanatory text and proof sketches to better convey intuition.
>
>
> **W2. Novelty relative to recent work**
>
> Great comment. Following the reviewer’s suggestion, we have carefully revised the text to better communicate our contributions. The updated text now more clearly highlights the specific novelty of OFM, which clarifies how our work builds upon, yet is distinct from, recent literature. Indeed it is subtle but in fact a very important difference.
>
>
> **W3. Theoretical assumption and scope**
>
> Thank you for raising this point. We now state explicitly in the main text that “we assume the target measure has full support on the same space as the reference GP.”  This absolute‑continuity assumption is standard in functional generative‑model literature (see Kerrigan et al., 2023; Lim et al., 2023; Seidman et al., 2023; Schneider et al., 2024), and we follow that convention here. For verifying whether processes lie in the Cameron-Martin space of the GP prior,  we refer to the detailed proof and discussion in Appendix C of Lim et al. (2023, see refs below). We have brought this point into the main text.
>
> Regarding flow matching, we emphasise that although the conditional path $\mu_t(\cdot\|z)$ is Gaussian, the marginal path obtained after integrating over $z$ is a mixture of Gaussians (weighted by the target data distribution) and therefore need not be Gaussian. This mirrors the finite‑dimensional case: in standard flow matching (Lipman et al., 2022, Eq. 11) and optimal‑transport flow matching (Tong et al., 2023, Thm. 2.1), a Gaussian conditional is paired with an unconstrained—and often highly non‑Gaussian—target distribution, such as images, audio, or video datasets. The resulting Gaussian mixture faithfully captures the complex structure of the target law. We have added an appendix section that reviews flow matching and elaborates this point.
>
>
>
> Reference :
>
> Kerrigan et al “ Functional Flow Matching” 2023
>
> Lim et al, “Score-based Diffusion Models in Function Space” 2023
>
> Seidman et al “Variational Autoencoding Neural Operators” 2023
>
> Schneider et al “An Unconditional Representation of the Conditional Score in Infinite Dimensional Linear Inverse Problems” 2024
>
> Lipman et al, “Flow Matching for Generative Modeling” 2022
>
> Tong et al, “Improving and generalizing flow-based generative models with minibatch optimal transport” 2023
>
> **W4. Empirical evaluation details**
>
> Following Reviewer 2aqv’s suggestion, we strengthened the 1D GP regression experiments by regenerating the training and test sets with more complex kernels—Gibbs / non‑stationary, Rational Quadratic kernel. OFM still outperforms all baselines on these cases as shown in the table (replied to Reviewer 2aqv)
>
> We acknowledge the current limitation in statistical robustness and plan a full multi‑seed validation in future work. For the baseline set‑up, Appendix A (Q5) explains why NDP is omitted from the quantitative tables and provides a detailed theoretical comparison instead in Appendix N
>
> On computational cost, the revised draft includes wall‑clock and peak‑memory numbers, together with a discussion of practical mitigations—e.g. mixed‑precision FNO kernels (Tu et al., 2023) or low‑step ODE solvers such as DPM‑Solver (Lu et al., 2022)—that can substantially reduce GPU time and memory.
>
> Reference.
>
> Tu et al, “Guaranteed Approximation Bounds for Mixed-Precision Neural Operators” 2023
>
> Lu et al, “DPM-Solver: A Fast ODE Solver for Diffusion Probabilistic Model Sampling in Around 10 Steps” 2022.
>
>
>
> **W5. Lack of the ablation in main text**
>
> To address this concern, we have reordered several figures for a clearer narrative flow and inserted a new ablation table in the main body.
>
>
> **W6. Evaluation on realistic tasks**
>
> Thank you for this valuable suggestion. We agree that this is a critical direction for future work. We intend to collaborate with domain experts to build a comprehensive benchmark of real-world datasets. This benchmark will specifically include challenging settings with non-i.i.d. and non-Gaussian noise. Developing this benchmark and extending our framework accordingly is a significant undertaking that warrants its own dedicated investigation. We are confident that this future work will enable more rigorous validation and help spur progress across many scientific and engineering fields.
>
>
> **Q1. Discuss the limitation in the main body rather than in the Appendix N**
>
> Thanks for the suggestion, we have moved the discussion of the model's limitations from the appendix into the main text in the updated draft to ensure it is more prominent.

---

> > ### Comment · Reviewer_sYHe · 2025-08-04
> >
> > I thank the authors for their response and for well addressing my comments. I keep my positive rating.

---

### Official Review · Reviewer_KpLg · 2025-07-02

**Clarity:** 2
**Significance:** 2
**Originality:** 3
**Rating:** 4
**Confidence:** 3

**Summary:**

The paper introduces a novel method, Operator Flow Matching (OFM), for learning stochastic processes. The approach consists of two key phases. First, it learns a neural operator that transforms samples from a Gaussian Process (GP) into samples from the target data distribution. Then, this learned operator is used to generate posterior samples given noisy observations. To address scalability and computational efficiency, the authors incorporate Stochastic Gradient Langevin Dynamics (SGLD) into the training procedure.

**Questions:**

Besides the questions raised in the weaknesses, I also wonder how Eq. 16 is computed in practice and why its complexity is claimed to be $O(m)$. A more detailed explanation would help clarify this point.

Additionally, the proposed method appears to align well with the denoising literature. It would be interesting to see how the flow matching framework performs on denoising tasks, which could further demonstrate its versatility and practical value.

**Ethical Concerns:**

["NO or VERY MINOR ethics concerns only"]

**Final Justification:**

My concerns are resolved. I have read the rebuttal responses and other reviews and made the final decision.

**Limitations:**

The authors have discussed the limitations in the Appendix, and the identified issues are common in both the Neural Process (NP)and the operator learning literature.

**Quality:**

2

**Strengths And Weaknesses:**

Strengths:
1. The paper is the first to extend flow matching to stochastic process learning via operator learning.
2. By leveraging a learned operator, the method can directly model data functions, enabling access to the exact posterior distribution over functions.
3. The approach is validated on several synthetic GP datasets and demonstrates improved performance compared to existing methods.

Weaknesses:
1. The paper lacks mathematical rigor and clarity in presentation.
a) It is not immediately clear that Eq. 14 is a solution to Eq. 11, nor is this connection explicitly stated. The authors are encouraged to include propositions or theorem-style statements to improve readability. Each phase should clearly state the optimization problem and how it connects to the method.
b) In line 230, the authors write: "For evaluating ... on the right-hand side of Eq. 18, we can efficiently calculate it with the likelihood estimation tool described above." It is unclear which likelihood estimation tool is being referenced. The statement should be made more precise.

2. The paper lacks a high-level explanation of flow matching, making it difficult to understand the intuition behind why this method might outperform alternatives like Neural Processes (NP). A conceptual overview of how flow matching contributes to the method would improve accessibility.

3. While an ablation study is included, it only focuses on the mini-batch size for optimal transport. However, the method involves other important hyperparameters. A broader sensitivity analysis would improve the robustness of the empirical evaluation.

4. The paper only evaluates on synthetic GP datasets. Real-world datasets commonly used in Neural Process literature—such as MNIST and CelebA for image completion—are missing. Including such benchmarks would strengthen the empirical impact (significance) of the work.

I would be inclined to raise my score if the clarity issues are addressed and additional experiments on real-world datasets and hyperparameter sensitivity are included.

---

> ### Author Rebuttal · Authors · 2025-07-31
>
> We appreciate the reviewer’s thoughtful feedback. Our responses below seek to clarify all concerns and potential misunderstandings
>
>
> **W1. The paper lacks mathematical rigor and clarity in presentation. a) it is not immediately clear that Eq. 14 is a solution to Eq. 11. b) unclear likelihood estimation tool being referenced**
>
> Thank you for these suggestions. Indeed, this paper presents a rigorous framework that combines several diverse topics to tackle the important problem of learning stochastic processes. As Reviewer sYHe also noted, the main text length limit of the paper has made it particularly challenging to communicate the relevant details from each of these fields. Nonetheless, we have tried our best to address this issue as follows. To further improve clarity for readers from diverse backgrounds,  we have (i) added an appendix section covering neural operators (operator learning), flow matching, Gaussian measures on function spaces, and the Cameron–Martin theorem to provide readers with essential background; and (ii) included additional explanatory text and theorem-style statements  to better convey intuition.
>
>
> To address the reviewer’s specific concerns:
>
> a)  To clarify, Eq. 11 is the training objective, which requires an explicit form for  $\mathcal{G}_t(u_t|z)$. Eq. 14 is then used to provide the explicit $\mathcal{G}_t(u_t|z)$ based on (derived from) the constructed probability path. We provided the full proof and derivation for Eq.14 in Appendix G (as mentioned in line 177, above Eq 14).  To further avoid any confusion, we have added a proposition before Eq. 14 to clarify this exactly in the updated draft.
>
>
> b) The phrase "the likelihood estimation tool described above" refers to the Hutchinson trace estimator discussed in Section 3.2. We have revised the text to remove this ambiguity.
>
>
> **W2. The paper lacks a high-level explanation of flow matching**
>
> To address this, we have added a dedicated section to the appendix providing a background and conceptual overview of flow matching.
>
> In brief, flow matching is a state-of-the-art generative paradigm that learns to transport samples from a simple distribution to a complex data distribution. Its strong performance and scalability have led to its adoption in highly challenging domains, including recent large-scale models for video generation, in-context image generation (e.g., FLUX.1 Kontext, 2025), and protein ensemble generation (Jing et al., 2024).  We selected flow matching as the foundation for our prior learning due to its numerous advantages. It has deep connections to physics (continuity equation) and offers simulation-free training with a simple learning objective. This generally leads to faster training and sampling compared to diffusion models, while also being fully invertible and highly scalable.
>
>
>
> Reference:
>
> “FLUX.1 Kontext: Flow Matching for In-Context Image Generation and Editing in Latent Space” 2025
>
> Jing et al, “AlphaFold Meets Flow Matching for Generating Protein Ensembles” 2024
>
>
> **W3. While an ablation study is included, a broader sensitivity analysis would improve the robustness of the empirical evaluation**
>
> We thank the reviewer for this constructive suggestion and agree that a broad sensitivity analysis would further strengthen our empirical evaluation. Given the extensive scope of this paper, which already spans multiple fields and involves a two‑phase framework with many interacting components, we prioritized demonstrating the core viability and performance of our approach across different tasks. Therefore, while we acknowledge its importance, a comprehensive sensitivity analysis across all components was beyond the scope of this initial investigation. We consider this a valuable and important direction for future work.
>
> **W4. The paper only evaluates on synthetic GP datasets, real-world dataset such as MNIST and CelebA are missing**
>
> We appreciate the opportunity to clarify our experimental evaluation.
>
> While our paper features GP datasets, the evaluation also includes several highly non-Gaussian benchmarks: $\textbf{2D Navier-Stokes}$, high-resolution simulations of $\textbf{black holes}$, and the $\textbf{MNIST-SDF}$ dataset (signed‑distance functions extracted from MNIST digits).
> For these non-Gaussian functional datasets, the standard likelihood metric is not applicable for OFM since the constant evidence term is intractable (as detailed in Appendix A, Q5), so we provide qualitative comparisons. These visualizations show that posterior samples from OFM align closely with the ground truth and preserve key features (e.g. density and swirling patterns of the blackhole) far better than those from baselines.
>
> To further strengthen the empirical section, and following Reviewer 2aqv’s suggestion, we have also added GP regression tasks using more complex kernels (Gibbs (non-stationary) kernel and Rational Quadratic kernel (RQK)). As shown in the following table, these new results confirm that OFM consistently outperforms all comparative methods across all metrics.
>
> | **Algorithm ↓ / Metric →** | **GP‑Gibbs&nbsp;SMSE** | **GP‑Gibbs&nbsp;MSLL** | **GP‑RQK&nbsp;SMSE** | **GP‑RQK&nbsp;MSLL** |
> |---|---|---|---|---|
> | NP | $4.0 \cdot 10^{-1}$ | $7.1 \cdot 10^{-1}$ | $5.8 \cdot 10^{-1}$ | $2.2 \cdot 10^{0}$ |
> | ANP | $3.5 \cdot 10^{-1}$ | $6.2 \cdot 10^{-1}$ | $2.7 \cdot 10^{-1}$ | $2.1 \cdot 10^{0}$ |
> | ConvCNP | $3.4 \cdot 10^{-1}$ | $1.1 \cdot 10^{-1}$ | $2.2 \cdot 10^{-1}$ | $6.9 \cdot 10^{-1}$ |
> | DGP | $4.3 \cdot 10^{-1}$ | $8.6 \cdot 10^{-1}$ | $2.4 \cdot 10^{-1}$ | $1.2 \cdot 10^{0}$ |
> | DSPP | $4.2 \cdot 10^{-1}$ | $7.1 \cdot 10^{-1}$ | $2.5 \cdot 10^{-1}$ | $4.2 \cdot 10^{-1}$ |
> | OpFlow | $3.1 \cdot 10^{-1}$ | $6.9 \cdot 10^{-1}$ | $2.9 \cdot 10^{-1}$ | $5.0 \cdot 10^{-1}$ |
> | **OFM&nbsp;(Ours)** | $\mathbf{2.9 \cdot 10^{-1}}$ | $\mathbf{8.7 \cdot 10^{-2}}$ | $\mathbf{2.1 \cdot 10^{-1}}$ | $\mathbf{9.4 \cdot 10^{-2}}$ |
>
> **Q1, how Eq. 16 is computed in practice and why its complexity is claimed to be O(m)**
>
> The divergence term in Eq. 16, which is the trace of a Jacobian, is computationally expensive to calculate exactly. We compute it in practice using the Hutchinson trace estimator, a standard, unbiased technique for neural ODE, first used by Grathwohl et al. (2018), which requires only one reverse‑mode autodiff pass through the network, so the time complexity is linear w.r.t cardinality O(m). Below is the PyTorch‑style pseudocode used in our implementation; the full script is included in the supplementary ZIP.
>
> ```
> # samples: target sample repeated n_noise times along the batch dimension.
> # choose Hutchinson noise
>
> if hutchinson_type == "Gaussian":
>     noise = torch.randn_like(samples)
> elif hutchinson_type == "Rademacher":
>     noise = (torch.randint_like(samples, 0, 2).float() * 2 - 1)
>
> with torch.enable_grad():
>     samples = samples.detach().requires_grad_(True)
>
>     # time condition
>     t_vec = torch.full((samples.size(0),), t, device=samples.device)
>
>     # drift gθ(t, x)
>     drift = model(t_vec, samples)
>
>     # Hutchinson estimate of divergence
>     f_eps = torch.sum(drift * noise)
>     grad = torch.autograd.grad(f_eps, samples,
>                                retain_graph=True)[0]
>     div_estimate = torch.sum(grad * noise,
>                              dim=tuple(range(1, grad.dim())))
> ```
>
>
> Reference:
>
> Grathwohl et al., "FFJORD: Free-form Continuous Dynamics for Scalable Reversible Generative Models." 2018.
>
> **Q2. It would be interesting to see how the flow matching framework performs on denoising tasks**
>
> This is a good point.  We discussed the technical connections between these two approaches in Appendix A (Q3). While we provide this  comparison, we agree that a comprehensive empirical evaluation on denoising-specific tasks is a valuable next step, which we leave for future work

---

> > ### Comment · Reviewer_KpLg · 2025-08-05
> >
> > I have read the responses. My concerns are resolved. The high-level explanation helps a lot. I will raise my score.

---

### Official Review · Reviewer_Ezjy · 2025-07-03

**Clarity:** 4
**Significance:** 3
**Originality:** 2
**Rating:** 5
**Confidence:** 2

**Summary:**

The authors introduce a method to perform flow matching in function spaces relying on a generalization of modelling functions to functionals. They demonstrate the applicability of the proposed approach to academic examples.

**Questions:**

**Q1: Scalability** The authors mention, their method is limited to low-dimensional settings and, so far, to problems where a simulator is available for unlimited data generation. Can you comment on which theoretical or practical advancements would be needed to overcome these challenges and apply the OFM method to real-world problems?

**Ethical Concerns:**

["NO or VERY MINOR ethics concerns only"]

**Final Justification:**

The paper contains a comprehensive introduction and simulation analysis of the proposed method. The method itself is a natural extension of flow matching and seems promising to me, and the authors discuss its limitations appropriately. Although the paper is very dense and reading the long appendix is almost mandatory to fully appreciate the work, it is well written and structured, and I do not see this as a reason to reject (as reviewer KpLg).

**Limitations:**

Limitations are discussed, but should be included in the main text.

**Paper Formatting Concerns:**

The paper is well written.

**Quality:**

3

**Strengths And Weaknesses:**

## Strengths
**S1: Interesting Generalization** The studied problem of generalizing flow matching to functionals is interesting and novel. It addresses a common limitation of CNP that can be of relevance in practice. The derivations seem correct to me.

**S2: Insights into the model** The experiments are described in detail, and the appendix provides comprehensive material on the analysis and comparison to other methods.

## Weaknesses

**W1:  Important insights are pushed to the appendix.** I appreciate the detailed appendix, but limitations or a short introduction on the main neural network baseline, the ConvCNP, should not be pushed to the appendix.

**W2: Excessive Data Requirements** The main strength of using GP and Bayesian methods in general is their performance in low-data regimes. The data requirements seem excessive for the low-dimensional examples provided. I read the limitation section in the appendix, but IMO it should be part of the main paper, as neither the data requirement nor the limitation to low-dimensional settings are mentioned in the main text.

---

> ### Author Rebuttal · Authors · 2025-07-31
>
> We appreciate the reviewer’s valuable comments and positive take on our work. We provide our feedback as follows.
>
> **W1. Important insights are pushed to the appendix**.
>
> Thanks for the suggestion, we have moved the overview of ConvCNP from the appendix into the main text so that readers can understand its role without consulting supplementary material.
>
> **W2. Excessive Data requirements**
>
> This is an important observation. We now make the trade-off explicit in the main text by adding:
>  “Although our approach attains state-of-the-art accuracy, it relies on substantially larger training datasets than are customary for GP-based models.”
>
>
> **Q1. Issues related to Scalability**
>
> The primary bottleneck is the general difficulty of learning operators for functions defined on high-dimensional domains (e.g., 3D space plus time). This is a foundational challenge in the operator learning field, which is still evolving to address both computational limitations and dataset availability
>
> Applying OFM to more complex problems will require key advancements on two fronts. First is the continued development of more scalable operator architectures that are both memory- and compute-efficient in higher dimensions. Second, and equally important, is the creation of large-scale, curated datasets for high-dimensional scientific problems, which are essential for training these more advanced operators.
>
> It's important to emphasize, however, that the current 1D and 2D settings are not merely toy problems; they encompass many critical and valuable real-world applications, such as time-series forecasting, audio processing, and tomography. Our work provides a robust foundation for these domains, while extending the framework to higher dimensions remains a key future goal dependent on these broader advances in the field.

---

> > ### Comment · Reviewer_Ezjy · 2025-08-04
> >
> > Thank you for your response and for addressing my comments. I will keep my positive rating.

---

### Official Review · Reviewer_2aqv · 2025-07-03

**Clarity:** 2
**Significance:** 3
**Originality:** 3
**Rating:** 5
**Confidence:** 3

**Summary:**

The paper proposes Operator Flow Matching (OFM), an extension of (optimal transport) conditional flow matching (CFM) to stochastic processes using neural operators. In a nutshell, the authors show that the vector fields in the CFM objective can be replaced with neural operators, thus generalising CFM to (infinite-dimensional) function spaces. This allows for evaluating the posterior probability of observations under any stochastic process as well as efficiently sampling from the process.

The authors evaluate their method on a range of experiments from simple synthetic 1D Gaussian and non-Gaussian data, where the true posterior is available, to more advanced data from simulations and MNIST. It shows promising performance, although at a high computational budget.

**Questions:**

**Suggestions**
1. My main suggestion would be to work on the clarity of the paper. For instance, it would have helped me to have a basic introduction to neural operators and (conditional) flow matching in the appendix. Also, I would suggest moving the technical details of the main paper to the appendix, and focusing the main paper on building intuition for your model and how to use it, perhaps with some proof sketches if necessary.
2. The figures and tables in the experiments section are placed rather chaotically, and table 1 is never mentioned, which is all somewhat distracting. It would help the reader if figures and tables were placed on the same page as where they are mentioned.

**Questions**
1. You mention that you extend neural operators to map between collections of points. Can you clarify how this is different from a standard flow?
2. In Eq. (11), what is the argument for replacing the suprema in Eq. (8) with expectations? When is this valid?
3. The OFM posterior samples in figures 8, 10, and 11 look quite noisy. Do you have ideas as to why this could be and how one might mitigate it?
4. For the GP experiments, the training set seems to be constructed from samples from the same GP (i.e., with fixed hyperparameters) that was also used for the test sample. This seems like an unrealistic setting in general. Is OFM able to learn from samples from GPs with different hyperparameters? If so, did you try that?
5. A minor question regarding the 1D GP experiment to make sure I understand the setup correctly: you write in appendix K that you sample the training set at 256 input locations, but table 3 says that you use a batch size of 1024 during training? Is this a mistake?

**Ethical Concerns:**

["NO or VERY MINOR ethics concerns only"]

**Final Justification:**

The authors have addressed my questions and concerns, and assuming the paper is updated to improve its clarity, I think it deserves to be accepted.

**Limitations:**

Yes.

**Paper Formatting Concerns:**

None noticed.

**Quality:**

3

**Strengths And Weaknesses:**

**Strengths**
1. The paper combines two recent and promising techniques, flow matching and neural operators. It appears original and should be of broad interest to the NeurIPS community.
2. The model itself is rigorously developed.
3. The model appears to work well in practice, despite some computational concerns (see weaknesses).
4. I could see this work forming the basis for many follow-up papers, so it could become quite significant.


**Weaknesses**
1. The main weakness of the paper, to me, is its clarity. It is very advanced material, which the authors are aware of and have tried to address by adding an FAQ and additional details to the appendix. While these are helpful, the main paper is still very detailed, and the sentences are long and dense, which makes the work difficult to understand.
2. As the authors themselves point out, the model is computationally heavy and is currently limited to low-dimensional data (the authors experiment on just 1D and 2D data).
3. The experimental section is a little weak, with many of the results deferred to the appendix. Some of the results are purely qualitative. The only quantitative experiments are some simple 1D and 2D synthetic datasets, which should at least have been extended to more complex examples (e.g., more complex GP kernels would be easy to test).

---

> ### Author Rebuttal · Authors · 2025-07-31
>
> We appreciate the reviewer’s valuable comments and positive take on our work. We provide our feedback as follows
>
> **Weakness 1 & Suggestion 1. The presentation is dense and contains very advanced material. The suggestion is to add a basic introduction to neural operators and flow matching in the appendix, focusing the main paper on building intuition for the model and its usage.**
>
> Thank you for these excellent suggestions. Indeed, this paper presents a rigorous framework that combines several rather diverse topics to tackle the important problem of learning stochastic processes. As Reviewer sYHe also noted, the main text length limit of the paper has made it particularly challenging to communicate the relevant details from each of these fields. Nonetheless, we have tried our best to address this issue as follows.
>
> To further improve clarity for readers from diverse backgrounds, we have (i) added an appendix section covering neural operators (operator learning), flow matching, Gaussian measures on function spaces, and the Cameron–Martin theorem to provide readers with essential background; and (ii) included additional explanatory text and proof sketches to better convey intuition.
>
> **Weakness 3. The experiment section is a little weak, add more complex examples (e.g., more complex GP kernels would be easy to test.**
>
> Following your suggestion, we have brought the important experimental findings from the appendix into the main text, and have subsequently expanded our 1D Gaussian Process (GP) experiments with more complex kernels for generating training datasets and regression tasks. Specifically, we now include results for the Gibbs (non-stationary) kernel and Rational Quadratic kernel (RQK) , As shown in the updated table below, our OFM model consistently outperforms all baselines on these tasks.
>
> | **Algorithm ↓ / Metric →** | **GP‑Gibbs&nbsp;SMSE** | **GP‑Gibbs&nbsp;MSLL** | **GP‑RQK&nbsp;SMSE** | **GP‑RQK&nbsp;MSLL** |
> |---|---|---|---|---|
> | NP | $4.0 \cdot 10^{-1}$ | $7.1 \cdot 10^{-1}$ | $5.8 \cdot 10^{-1}$ | $2.2 \cdot 10^{0}$ |
> | ANP | $3.5 \cdot 10^{-1}$ | $6.2 \cdot 10^{-1}$ | $2.7 \cdot 10^{-1}$ | $2.1 \cdot 10^{0}$ |
> | ConvCNP | $3.4 \cdot 10^{-1}$ | $1.1 \cdot 10^{-1}$ | $2.2 \cdot 10^{-1}$ | $6.9 \cdot 10^{-1}$ |
> | DGP | $4.3 \cdot 10^{-1}$ | $8.6 \cdot 10^{-1}$ | $2.4 \cdot 10^{-1}$ | $1.2 \cdot 10^{0}$ |
> | DSPP | $4.2 \cdot 10^{-1}$ | $7.1 \cdot 10^{-1}$ | $2.5 \cdot 10^{-1}$ | $4.2 \cdot 10^{-1}$ |
> | OpFlow | $3.1 \cdot 10^{-1}$ | $6.9 \cdot 10^{-1}$ | $2.9 \cdot 10^{-1}$ | $5.0 \cdot 10^{-1}$ |
> | **OFM&nbsp;(Ours)** | $\mathbf{2.9 \cdot 10^{-1}}$ | $\mathbf{8.7 \cdot 10^{-2}}$ | $\mathbf{2.1 \cdot 10^{-1}}$ | $\mathbf{9.4 \cdot 10^{-2}}$ |
>
>
>
> **Suggestions 2. The figures and tables in the experiments section are placed rather chaotically, and table 1 is never mentioned, which is all somewhat distracting. It would help the reader if figures and tables were placed on the same page as where they are mentioned.**
>
> Thanks for pointing out ways to improve the flow of the paper. To address this, we have moved Table 1 to the appendix and have carefully reordered and resized several figures to place them closer to where they are discussed in the text. We also respectfully note that Table 1 was in fact referenced in Section 3.1 (Line 190).
>
> **Q1. You mention that you extend neural operators to map between collections of points. Can you clarify how this is different from a standard flow?**
>
> Thank you for this question. A standard flow‑matching model learns a transport map between two distributions defined on a fixed grid (e.g., a pixel lattice). Consequently, it can only generate samples at that specific resolution
>
> In contrast, OFM learns a map between two stochastic processes in function space. This endows it with several properties that standard flow matching lacks:
>
> (i) Resolution agnosticism. OFM learns a transport map between two distributions defined on any given collection of points, without regard for the number of points, or their locations in the domain. This enables capabilities like zero-shot generation without retraining.
>
> (ii) Stochastic process consistency: OFM respects the metric of the underlying space, ensuring that points close to each other in the input domain have appropriately correlated values in the output distribution. This is part of learning a valid stochastic process, which also satisfies crucial theoretical properties like the Kolmogorov extension theorem (i.e, consistency under marginalization)
>
> (iv) Convergence to a Continuous Function: As the collection of query points becomes denser within the domain, the output of OFM converges to the underlying continuous function.
>
> (v) Backwards Compatibility: When evaluated on a fixed set of points, OFM recovers the behavior of a conventional (finite-dimensional) flow matching up to a linear transformation.
>
>
> **Q2. In Eq (11). What is the argument for replacing the suprema in Eq. (8) with expectations? When is this valid?**
>
> The supremum in Eq. (8) represents a worst-case error over the entire function space, which is computationally intractable to optimize directly. We therefore relax this hard constraint by replacing the supremum with an expectation, as formulated in Eq. (11). This is a common empirical consideration.
>
> Instead of minimizing the worst-case error, our objective becomes minimizing the tractable average-case error across the distribution. Minimizing the error on average provides a strong practical incentive for the model to perform well across the entire function space, thereby effectively reducing the worst-case error. Such replacement is always valid under the weaker goal. The validity of this approach as a tractable proxy is further confirmed by our empirical results, which show it successfully guides the model to learn the intended functional mapping (Appendix K). In the revised manuscript, we have also added text to explicitly acknowledge the limitations of this relaxation.
>
> **Q3. OFM posterior samples in figure 8, 10, 11 look noisy, any ideas why this could be and how one might mitigate it**
>
> Posterior sampling in our framework relies on the unbiased Hutchinson trace estimator: we average the score under $n_{noise}$ (Appendix M) Gaussian noise draws to approximate the log‑determinant term. While the estimator is inexpensive in time, it is memory‑bound because all noise realizations must be processed in parallel. Owing to GPU memory limits we used only  $n_{noise}$=8 in Figures 8, 10, 11, which introduces noticeable variance and in turn yields larger stochastic gradients in each SGLD step (Algorithm 1), manifesting as visual noise..
>
> There are several clear paths to mitigate this:
>
>  (i) Carefully adjusting the step size ($\eta_t$) and temperature ($T$) in the SGLD algorithm to prevent overly large gradient updates.
>
> (ii) Running the trace estimator across multiple GPUs or using mixed‑precision FNO for memory saving (Tu et al, 2023).
>
> (iii) Using more efficient ODE solvers. Our current implementation uses a high-order ODE solver (dopri5) that requires over 100 steps for sampling. Switching to a more modern and efficient solver, such as the DPM-Solver (Lu et al., 2022), could potentially reduce the number of evaluations, freeing up significant GPU resources that could be allocated to increasing $n_{noise}$.
>
> Reference:
>
> Tu et al, “Guaranteed Approximation Bounds for Mixed-Precision Neural Operators” 2023
>
> Lu et al, “DPM-Solver: A Fast ODE Solver for Diffusion Probabilistic Model Sampling in Around 10 Steps” 2022.
>
>
> **Q4. The GP experiments, the training set seems to be constructed from samples from the sample GP. Is OFM able to learn from samples from GPs with different hyperparameter**
>
> There may be a slight misunderstanding of our experimental setup, and we appreciate the opportunity to clarify.  OFM is designed to learn the distribution of a single target stochastic process (GP or non-GP). The training set and test examples must therefore come from the same underlying process.
>
> The model first learns this specific process during the prior learning phase. It can then perform inference for that learned process, but it cannot generalize to a new process without being retrained.
>
> **Q5. Appendix K that you sample the training set at 256 input locations, but table 3 says use a batch size of 1024 during training**
>
> Thank you for the question. These two numbers refer to distinct stages of our framework and don't contradict one another. The batch size of 1024 in Table 3 specifies the number of independent function samples used in each training step during the prior learning phase. In contrast, the 256 input locations mentioned in Appendix K refers to the number of points used to discretize a single underlying function during the posterior sampling phase. In short, 1024 is the number of functions in a training batch, while 256 is the resolution at which we evaluate an individual function.

---

> > ### Comment · Reviewer_2aqv · 2025-08-05
> >
> > Dear authors,
> >
> > Thank you very much for your rebuttal. It cleared up many things for me and addressed my questions and concerns. Assuming that the updates to the text, which you have mentioned in your rebuttal to me and the other reviewers, are incorporated, I will increase my score.

---

### Decision · Program_Chairs · 2025-09-17

**Decision:**

Accept (spotlight)

**Comment:**

The reviewers are uniformly positive about this paper, which they agree is interesting, and original, and would be a useful addition to the literature. The biggest concern was about how dense the presentation was, on the one hand, this is understandable given the material, at the same time, the authors made some proposals to alleviate the burden on the reader. Please go over these (along with the rest of the discussion with the reviewers) while preparing the camera-ready version.